# Kidney intercalated cells are phagocytic and acidify internalized uropathogenic *Escherichia coli*

Vijay Saxena [1,5 ✉], Hongyu Gao[2], Samuel Arregui [1], Amy Zollman[3], Malgorzata Maria Kamocka[3], Xiaoling Xuei[2], Patrick McGuire[2], Michael Hutchens [4], Takashi Hato [3], David S. Hains [1,5] & Andrew L. Schwaderer [1,5 ✉]

Kidney intercalated cells are involved in acid-base homeostasis via vacuolar ATPase expression. Here we report six human intercalated cell subtypes, including hybrid principal-intercalated cells identified from single cell transcriptomics. Phagosome maturation is a biological process that increases in biological pathway analysis rank following exposure to uropathogenic *Escherichia coli* in two of the intercalated cell subtypes. Real time confocal microscopy visualization of murine renal tubules perfused with green fluorescent protein expressing *Escherichia coli* or pHrodo Green *E. coli* BioParticles demonstrates that intercalated cells actively phagocytose bacteria then acidify phagolysosomes. Additionally, intercalated cells have increased vacuolar ATPase expression following in vivo experimental UTI. Taken together, intercalated cells exhibit a transcriptional response conducive to the kidney's defense, engulf bacteria and acidify the internalized bacteria. Intercalated cells represent an epithelial cell with characteristics of professional phagocytes like macrophages.

[1] Indiana University School of Medicine, Department of Pediatrics, Division of Nephrology, Indianapolis, IN, USA. [2] Indiana University School of Medicine, Department of Medical & Molecular Genetics, Indianapolis, IN, USA. [3] Indiana University School of Medicine, Department of Medicine, Division of Nephrology, Indianapolis, IN, USA. [4] Oregon Health and Science University, Department of Anesthesiology & Perioperative Medicine, Portland, OR, USA. [5] These authors contributed equally: Vijay Saxena, David S. Hains, Andrew L. Schwaderer. ✉email: visaxena@iu.edu; schwadea@iu.edu

The kidney-collecting tubule is the terminal location for acid–base regulation[1]. The collecting tubule consists of principal cells and intercalated cells (ICs). Principal cells (PCs) mediate salt and water resorption and are characterized by cytoplasmic aquaporin 2 (AQP2)[2]. ICs help maintain acid–base homeostasis and have three described subtypes, type A (A-IC), type B (B-IC), and nonA–nonB[2,3]. A-ICs secrete protons via apical vacuolar H-ATPase (V-ATPase) and regenerate bicarbonate via the basilar chloride/bicarbonate ($Cl^-/HCO_3$) transporter, anion exchange 1 (AE1/band 3/SLC4A1)[1,4]. B-ICs secrete $HCO_3^-$ via an apical $Cl^-/HCO_3$ transporter, pendrin (SLC26A4), and express basolateral V-ATPase[2,3]. Rodent nonA–nonB ICs have apical pendrin and V-ATPase, but no AE1, and have been identified in the collecting duct and connecting tubule (CNT)[3]. Increasing evidence supports an innate immune role for ICs along with their traditional function of pH regulation. First, the majority of ten unrelated patients with distal renal tubular acidosis, a condition characterized by IC dysfunction, were reported to have a UTI history[5]. Second, uropathogenic *Escherichia coli* (UPEC), the organism isolated in 70% of acute pyelonephritis in males and 80% in females, selectively localizes to the cytoplasm of ICs[6,7]. Third, we identified that ICs express antimicrobial peptides (AMPs) such as ribonuclease 7 (RNASE7)[8–11]. These AMPs have direct activity against a range of pathogens, including bacteria, at baseline and/or in response to infection[8–11]. Fourth, we and others have independently reported that mice with abnormal IC development have increased susceptibility to urinary tract infections (UTIs) and/or pyelonephritis[12,13]. Fifth, ICs both sense and mediate inflammation[14]. Last, we have demonstrated innate immune gene expression in isolated murine ICs[15,16].

Understanding how ICs defend the urinary tract from infection is important because UTIs are frequently encountered. Over 50% of women experience a UTI during their lifetime, and UTIs account for 1–6% of all medical visits[17]. When bacteria ascend to the kidneys resulting in pyelonephritis, potential complications include hypertension, chronic kidney disease, and urosepsis[18,19]. An estimated 250,000 cases of pyelonephritis occur annually in the United States[17]. Multidrug resistance is emerging as a critical challenge regarding treatment of UTIs and other infections[20]. Therefore, the mechanisms ICs use to defend against ascending uropathogens might be therapeutic targets that could be exploited to develop kidney-specific pyelonephritis therapy. Because most IC studies involve animal models, a better understanding of human ICs will be needed to guide basic science studies into improved clinical care.

Because the kidney is a complex organ, mRNA profiles can be difficult to interpret. For example, the nephron has regional specialization with distinct tubular segments and cell subtypes. The kidney contains endothelial, epithelial, interstitial, and recruited inflammatory cells, all with distinct biological functions and gene expression patterns. Single-cell RNA sequencing (scRNA-seq) represents a rapidly advancing methodology to identify and profile distinct cell types at baseline or in response to stressors[21]. Here, we show using scRNA-seq technology that human kidney ICs consist of six different subsets that include three A-ICs, one hybrid IC–PC, one nonA–nonB IC, and one B-IC. RNA velocity analysis revealed transcriptional shift from PCs toward A-IC subsets upon UPEC exposure. Ingenuity Pathway Analysis revealed phagosome maturation as the key biological pathway in an A-IC subset upon UPEC exposure. Within this pathway, V-ATPases are the key involved components. We used intravital microscopy to visualize in vivo UPEC uptake by ICs and confirmed acidification of UPEC internalized in these ICs. Finally, a murine UTI model demonstrated increase in V-ATPase (Atp6v1b1) mRNA expression. This study will provide the foundation to explore and therapeutically manipulate V-ATPase expressing epithelial cells as bacterial phagocytes.

## Results

**Human ICs can be enriched via magnetic-activated cell sorting (MACS) with mast/stem cell growth factor receptor (CD117/c-KIT) coated magnetic beads.** Human kidney tissue, primarily from normal margins of kidney mass resections, was dissected into 2–4-mm pieces then overnighted to our lab in Dulbecco's Modified Eagle Medium (DMEM) on ice by the Cooperative Human Tissue Network [https://www.chtn.org]. We evaluated c-KIT as cell surface marker to sort human ICs because c-KIT has been successfully used for this purpose in mice[22]. c-KIT expression localized to human ICs based on the finding that c-KIT and the human IC marker V-ATPase, B1 subunit (gene ATP6V1B1), co-labeled the same cells in the human kidney section (Supplementary Fig. 1)[22,23]. Following magnetic sorting which included removal of CD45+ immune cells and dead cells; the enrichment of c-KIT-positive presumed ICs was demonstrated by ATP6V1B1 mRNA expression that increased by 8–12-fold in c-Kit-positive compared to the c-KIT-negative cells. Both SLC4A1 and SLC26A4 were variably enriched in ICs demonstrating that both A-IC and/or B-IC enrichment is possible using this methodology (Supplementary Fig. 2). The background of the kidney tissue used in this study is presented in Supplementary Table 1.

**Integrated cluster analysis identified six IC subtypes.** ScRNA-seq was completed on 1861 dissociated kidney cells that were enriched for ICs as described above. The enriched ICs were obtained from the normal margins of a single kidney mass resection. Quality control (QC) filtering was passed by 859 cells exposed to UPEC in vitro for 1 h and 1002 cells exposed to saline in vitro for 1 h. Seurat function analysis identified 12 clusters in the sorted cell preparation (Fig. 1a). The nine most conserved markers within each cluster are presented in Supplementary Figs. 3–14. Overall, IC subtypes accounted for 1066 (57%) of the 1861 cells. Six of the clusters represented IC subtypes. Specifically, we found three A-IC, one B-IC, one nonA–nonB-IC, and one hybrid PC–IC subtype(s). Kidney epithelial cell nomenclature recommended by Chen and colleagues, conserved makers that have been previously reported to be cell-type markers or conserved markers that differentiated one of the 12 clusters from others were used to assign a cell type to each cluster (Supplementary Table 2, Fig. 1b, c, Fig. 2)[24–28]. Expression levels of all genes and cluster-specific marker genes can be found at https://hpcwebapps.cit.nih.gov/ESBL/Database/IU-Data/Human-c-Kit-Sorted-Single-Cell-RNASeq.htm.

**A-IC subtype and hybrid PC–IC protein expression correlates with scRNA-seq results.** We validated key scRNA-seq findings, notably that SLC8A1 (also referred to as $Na^+/Ca^+$ exchanger 1 (NCX1)) is a marker for hybrid PC–ICs and that heat shock protein family A (Hsp70) member 1A (HSPA1A) along with early growth response 1 (EGR1) are markers that can differentiate A-IC subtypes (Fig. 3). Because we used the normal margins of kidney cancer resections, we validated that our SLC8A1, HSPA1A, and EGR1 kidney expression patterns were consistent to that seen in normal kidney tissue from the Human Protein Atlas available from http://www.proteinatlas.org (Supplementary Fig. 15)[29].

**Collecting duct cell subtype marker profiles previously demonstrated in murine kidney cells remain largely intact in human kidney cells.** Because it will be important to contextualize future murine model findings to human pathophysiology, we evaluated human IC expression of murine-collecting duct cell

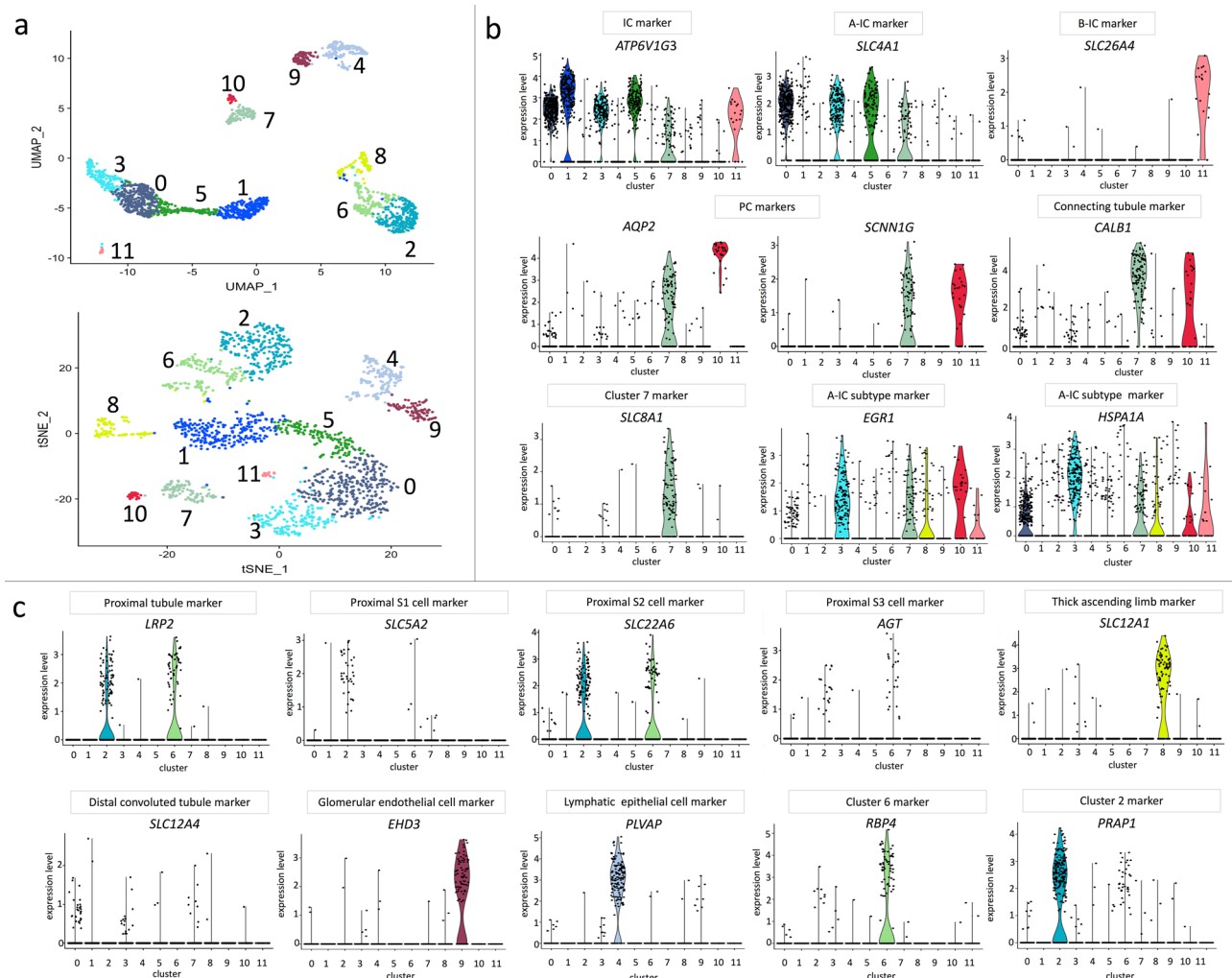

**Fig. 1 Human kidneys contain six subtypes of ICs and *SLC8A1* is highly selective to hybrid PC–ICs. a** Cell cluster analysis revealed that uniform manifold approximation and projection (UMAP) (top plot) and *t*-distributed stochastic neighborhood embedding (tSNE) (bottom plot) resulted in comparable identification of 12 clusters. **b** Expression of collecting duct marker genes is presented. Historical cell-specific markers were used to identify the cell type associated with each cluster. For some clusters, further classification was needed. *HSPA1A* and *EGR1* were used to differentiate clusters 1, 3, and 5. **c** Expression of cell marker gene expression for noncollecting duct kidney cells is presented. *SLC8A1*, *RBP4*, and *PRAP1* were used as markers for clusters 7, 6, and 2, respectively. The colors in the violin plots (**b**, **c**) correspond to the cluster color in the UMAP and tSNE plots (**a**).

markers identified previously in scRNA-seq studies performed by Chen et al.[22] and Ransick et al.[30]. A dot plot of the results is presented in Fig. 4. Murine-collecting duct cell-type markers are mostly conserved in human collecting duct cells.

**A-IC subtype A increases in relative percentage with 1 h of UPEC exposure.** The percentages of cells within clusters were consistent between exposure type with the exception of A-IC subtype A/cluster 0, which increased from 16.2 to 20.1%, and A-IC subtype C/cluster 5, which decreased from 9.4 to 6.6%, with saline vs. UPEC exposures, *P* = 0.03 (Table 1).

**Hybrid PC–IC cells change their RNA velocity away from PCs and toward A-ICs in response to 1 h of UPEC exposure.** The relative abundance of recently transcribed unspliced pre-mRNAs vs. mature spliced mRNA can be used to calculate the change in mRNA abundance, termed mRNA velocity[31]. A positive mRNA velocity in single-cell studies indicates that genes are being upregulated where a negative mRNA velocity means that genes are being downregulated[31]. The RNA velocity direction infers that a cell has a mRNA expression trajectory toward another cell

type[31,32]. The mRNA velocity in response to UPEC exposure is presented in Fig. 5. Of importance, hybrid PC–ICs change their RNA velocity away from PCs and toward ICs in response to UPEC. In addition, ICs maintain their transcription activity, whereas transcriptional activity becomes minimal in other kidney cell types.

**The kidney innate immune profile demonstrates some early single-cell expression changes following 1 h of UPEC exposure.** scRNA-seq expression patterns of select innate immune genes following UPEC vs. saline exposure for 1 h are presented in Supplementary Figs. 16 and 17. Key findings include high expression of the AMP adrenomedullin (*ADM*) in A-IC subtypes A and B, hybrid PC–ICs, and B-ICs. PCs do not express adrenomedullin at baseline but have significantly increased expression in response to UPEC exposure. Cytokine-inducible SH2-containing protein (*CISH*) and barrier to autointegration factor 1 (*BANF1*) expression is significantly induced in nonA–nonB ICs following UPEC exposure. Interleukin 18 (*IL18*), galectin 3 (*LGALS3*), beta defensin 1 (*DEFB1*), and signal transducer *CD24* lipocalin 2/neutrophil gelatinase-associated lipocalin (*LCN2/NGAL*) was only identified

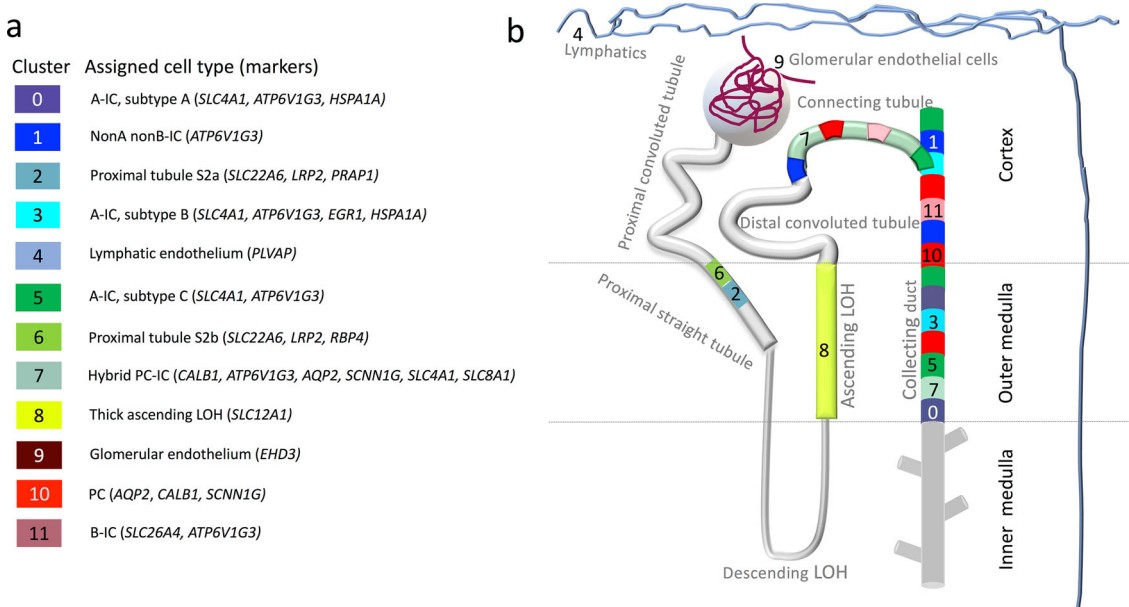

**Fig. 2 Cell types assigned to each cluster based on marker gene expression. a** The cell types and the gene markers from Fig. 1b, c used to assign the cell type are presented. The colors in the key correspond to the cluster colors from Fig. 1a. ICs were identified by *ATP6V1G3* (a subunit of V-ATPase) expression and consisted of clusters 0, 1, 3, 5, 11, and 12. A-ICs were identified by *SLC4A1* expression and consisted of 0, 3, 5, and 7. B-ICs were identified by *SLC26A4* expression that was limited to cluster 11. PCs were identified by *AQP2* expression which was prominent in clusters 7 and 10. While cluster 10 appeared to be traditional PCs, cluster 7 were hybrid PCs–ICs that expressed both *AQP2* and *SLC4A1*. *SLC8A1* was highly selective to hybrid PC–ICs that also expressed *CALB1*, indicating that these cells might originate in the connecting segment. There were three subtypes of A-ICs that were characterized by *HSPA1A* and *EGR1* expression: A-IC subtype A/cluster 0 expressed *HSPA1A*, A-IC subtype B/cluster 3 expressed both *HSPA1A* and *EGR1*, A-IC subtype C/cluster 5 expressed neither *HSPA1A* nor *EGR1*. **b** The expected location of the cell types within the kidney is presented.

in hybrid PC–ICs and only minimal toll-like receptor 4 mRNA expression was present. We did not identify expression of ribonuclease 7 (*RNASE7*) on scRNA-seq. We evaluated *RNASE7* mRNA expression in magnetically sorted pooled IC and non-IC kidney cells. *RNASE7* mRNA expression was identified in ICs from one out of four kidneys (Supplementary Fig. 18).

**The biological process of phagosome maturation is associated with ICs exposed to UPEC for 1 h.** The ten leading biological pathways associated with each cluster were ranked by *p* value as determined by Ingenuity Pathway Analysis. Three collecting duct-assigned clusters A-IC subtype A/cluster 0, A-IC subtype B/cluster 3, and A-IC subtype C/cluster 5 had differential biological pathway processes following UPEC exposure compared to saline (Fig. 6a–c). Phagosome maturation was the top ranked function in cells exposed to UPEC and saline in A-IC subtype C/cluster 5, moved from second to first position in A-IC subtype A/cluster 0 and from fourth to third highest position in A-IC subtype B/cluster 3. None of the remaining collecting duct cell subtypes (Supplementary Fig. 19) had top ten associated pathways that were different following UPEC exposure compared to saline. The gene expression profile that resulted in the phagosome maturation ranking for A-IC subtype A/cluster 0 is presented in Supplementary Table 3.

**IC *Atp6v1b1* mRNA expression increases in response to experimental pyelonephritis in vivo.** The Ingenuity phagosome maturation pathway highlighted V-ATPase as involved in ICs (Supplementary Table 3). To determine if V-ATPase is activated upon UPEC exposure, we used a murine UTI model. One hour post transurethral UPEC vs saline inoculation "IC reporter" mice, which have tdTomato (tdT, a red fluorescent protein variant) expression in ICs, were euthanized and ICs were enriched from

dissociated kidney cells. Enriched ICs had higher *Atp6v1b1* mRNA expression in mice with UPEC inoculation compared to saline control (Fig. 6d).

**ICs phagocytize UPEC in vivo.** We developed methodology to perfuse a single kidney tubule with UPEC in a live mouse with an intact kidney. Green fluorescent protein (GFP)-expressing UPEC aggregates microperfused into a single kidney tubule lumen of "IC reporter" mice were visualized flowing through tubules, selectively adhering to the luminal surface and being internalized by ICs (Fig. 7) during intravital microscopy.

**ICs acidify UPEC containing phagolysosomes in vivo and in vitro.** We also perfused tubules with pHrodo Green *E. coli* BioParticles that only fluoresce when acidified. Uptake and fluorescence of these bioparticles was only visualized during intravital microscopy in ICs (Fig. 8). Imaris segmentation analysis demonstrated that green fluorescence significantly increased over time in tdT$^+$ ICs within 2 tubules perfused with pHrodo Green *E. coli* BioParticles compared to a control tdT$^+$ tubule (Fig. 9a, b). When fluorescence was evaluated on a cellular basis, 12/16 (75%) tdT$^+$ cells demonstrated an increase in green fluorescence over time following tubular perfusion with pHrodo Green *E. coli* BioParticles (Fig. 9b). The linear regression results from each cell are presented in Supplementary Table 4. The live in vivo intravital imaging findings of phagocytosis and acidification of PHrodo Green *E. coli* BioParticles by ICs were validated using an in vitro flow cytometry assay. Murine kidney cell suspension from "IC reporter" mice in which ICs (CD45$^-$tdT$^+$) endogenously express tdT was exposed to pHrodo Green *E. coli* BioParticles versus cells with media alone. CD45$^+$ resident immune cells in cell suspension were gated as a positive control and CD45$^-$tdT$^-$ (non-ICs) cells were gated as a negative control. Comparing media alone to

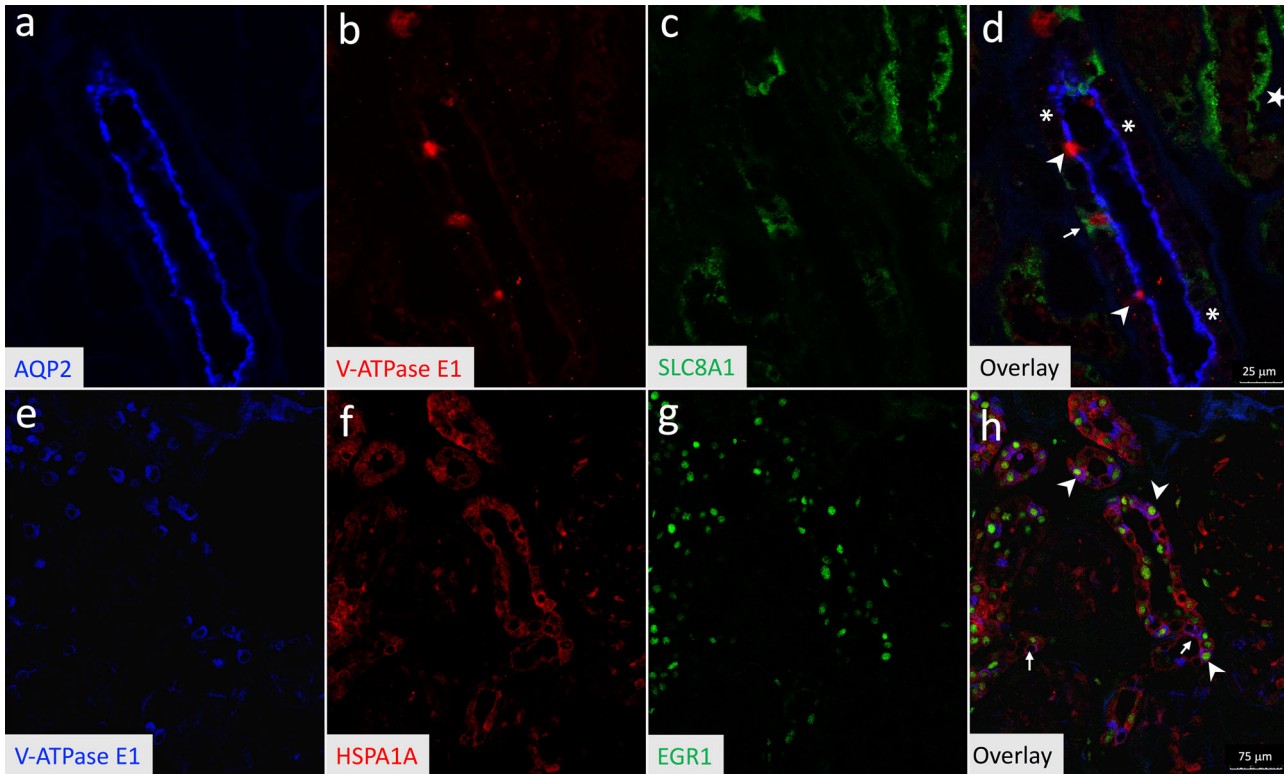

**Fig. 3 Hybrid PCs–ICs and IC subtypes are seen on fluorescent immunolabeling.** Triple-fluorescent immunolabeling was completed with the PC marker AQP2 (blue) (**a**), the IC marker V-ATPase E1 (red) (**b**), and proposed hybrid PC-IC cell marker SLC8A1 (green) (**c**), and images were obtained using confocal microscopy. The overlay (**d**) demonstrates a triple-labeled hybrid PC–IC cell (arrow), along with cells with only AQP2 labeling consistent with PCs (asterisks) and cells only with V-ATPase E1 labeling (arrowheads) consistent with ICs. Some tubules that did not express V-ATPase E1 or AQP2 had a majority of cells that were positive for SLC8A1 (star). Triple-fluorescent immunolabeling was also completed for A-IC subtypes with V-ATPase E1 (blue) (**e**), HSPA1A (red) (**f**), and EGR1 (green) (**g**). The confocal microscopy image overlay (**h**) contained cells with V-ATPase E1, HSPA1A, and EGR1 labeling consistent with A-IC subtype B (arrowheads) along with cells with HSPA1 and V-ATPase E1 but not EGR1 labeling consistent with A-IC subtype A (arrows). Representative images are presented from human kidney sections from two distinct individuals. Source data are provided as a Source Data File.

bioparticle exposure, green fluorescence indicative of bioparticle acquisition and acidification increased from 0.7 to 62.8% of CD45$^+$ cells. This finding demonstrates the expected bacterial uptake by professional phagocytes such as neutrophils. Interestingly, 22.2% of the ICs had uptake of green bioparticles compared to 5.6% of non-ICs (Supplementary Fig. 20), media control had uptake in 0.7% of ICs and 0% of non-ICs.

## Discussion

ICs are difficult to study, particularly in humans, because they comprise only 7% of kidney cells, consist of a range of subtypes, and do not retain their phenotype in culture[33]. Chen et al.[22] reported single-cell analysis on murine kidney cells and Lake et al. reported a single-nucleus RNA sequencing pipeline on human kidney cells[34,35]. These studies identified two subtypes of A-ICs. Here, we have developed methodology to perform single-cell sequencing on viable enriched human ICs. We enriched for viable ICs rather than sequencing all kidney cells to allow increased focus on this cell type. Our primary findings were (a) identification that hybrid PCs–ICs can switch their RNA velocity direction away from PCs and toward A-ICs in response to UPEC exposure, (b) at least three subtypes of A-ICs exist and their relative frequencies can shift in response to UPEC exposure, (c) phagosome maturation is a leading biological pathway in multiple A-IC subtypes and can increase in relative significance with UPEC exposure, and (d) ICs phagocytose and acidify UPEC in vivo during imaging of live mice.

ICs differentiate in response to acidosis by reversing their polarity[36,37]. ICs have been reported to arise from AQP2-expressing cells[38]. In 2017, Chen and colleagues identified double-positive cells that expressed Aqp2 along with Slc4a1 or Slc26a4 by murine single-cell profiling. These findings were validated by Park and colleagues the following year[22,35]. In 2019, we demonstrated by protein and mRNA immunolabeling that hybrid PC–ICs exist[15]. Specifically, 9% of murine ICs defined by V-ATPase B1 fluorescent immunolabeling also expressed Aqp2 mRNA defined by fluorescent in situ hybridization[15]. The present study defined a population of human kidney-collecting duct cells that express AQP2, ATP6V1G3, and SLC4A1 consistent with hybrid PC–ICs. On tSNE and UMAP plots, these hybrid cells clustered closest to PCs, but were distinct from traditional PCs based on IC marker expression. SLC8A1 expression was distinct to this cell type. Interestingly SLC8A1 is highly involved with epithelial-to-mesenchymal transition. Specifically, the absence of SLC8A1 transforms epithelial cell phenotypes via β-catenin-mediated destabilization of E-cadherin[39]. Here, we demonstrate that ICs differentiate in response to uropathogen exposure. Hybrid PC–IC cells change their RNA velocity, the time derivative of mRNA expression, away from PCs and toward A-ICs in response to UPEC. In mice, Ransick et al.[30] identified Slc8a1 expression primarily in the distal convoluted tubule and cells resembling PCs in the connecting tubule. On our confocal microcopy images evaluating kidney SLC8A1 expression, we found isolated cells that expressed SLC8A1 in AQP2-positive tubules (e.g., collecting ducts or connecting tubules). We also

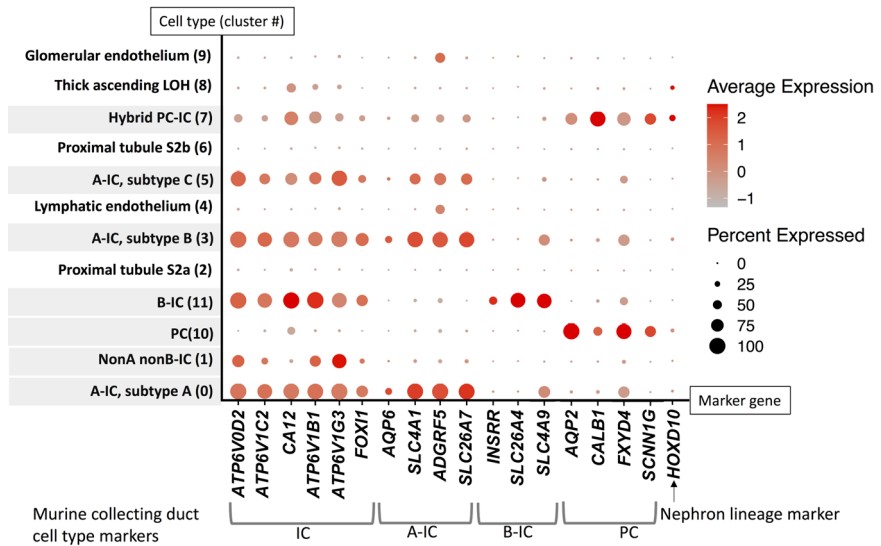

**Fig. 4 Murine-collecting duct cell-type marker expression patterns are generally conserved in human collecting duct cells.** IC markers include the D2, C2, B1, and G3 subunits of V-ATPase (*ATP6V0D2, ATP6V1C2, ATP6V1B1,* and *ATP6V1G3* respectively), carbonic anhydrase 12 (*CA12*), and forkhead box protein I1 (*FOXI1*). A-IC markers include aquaporin 6 (*AQP6*), *SLC4A1,* adhesion G protein-coupled receptor F5 (*ADGRF5*), and solute carrier family 26 member 7 (*SLC26A7*). B-IC markers include insulin receptor-related receptor (*INSRR*), *SLC26A4,* and solute carrier family 4 member 9 (*SLC4A9*). PC markers include *AQP2,* calbindin 1 (*CALB1*), *FXYD* domain containing ion transport regulator 4 (*FXYD4*), and *SCCN1G*. Homeobox D10 (*HOXD10*) is a nephron lineage marker. A dot plot of murine-collecting duct cell-type marker gene expression in human kidney cells is presented. Collecting duct cell types are shaded. Murine markers for ICs and A-ICs are expressed in all human ICs and A-ICs of the same subtypes. The murine B-IC marker *SLC4A9* is expressed in human A-IC subtypes A and B along with human B-ICs. Murine PC marker genes are primarily limited to PCs and hybrid PC–ICs in humans, except for *FXYD4,* which is also expressed in some A-IC subtypes. The majority of human collecting duct cells in this analysis were *HOXD10* negative, indicating ureteric origin. PCs and PCs–ICs were the cells that had the highest percentage of *HOXD10* positivity indicating derivation from the distal nephron. However, nephron-derived cells still represented a minority of PCs and PCs–ICs. Saline-exposed cell results are presented.

**Table 1 Differential cell type frequency in UPEC vs. saline exposed cells.**

| Cluster | Assigned cell type | Saline (n = 1002 cells) | UPEC (n = 859 cells) | P value |
|---|---|---|---|---|
| 0 | A-IC, subtype A | 162 (16.2%) | 173 (20.1%) | 0.03* |
| 1 | NonA nonB-IC | 152 (15.2%) | 119 (13.9%) | 0.43 |
| 2 | Proximal tubule S2a | 133 (13.3%) | 121 (14.1%) | 0.64 |
| 3 | A-IC, subtype B | 94 (9.4%) | 84 (9.8%) | 0.81 |
| 4 | Lymphatic endothelium | 95 (9.5%) | 72 (8.4%) | 0.42 |
| 5 | A-IC, subtype C | 94 (9.4%) | 57 (6.6%) | 0.03* |
| 6 | Proximal tubule S2b | 86 (8.6%) | 64 (7.5%) | 0.39 |
| 7 | Hybrid PC–IC | 54 (5.4%) | 57 (6.6%) | 0.28 |
| 8 | Thick ascending LOH | 57 (5.7%) | 44 (6.1%) | 0.61 |
| 9 | Glomerular endothelium | 52 (5.2%) | 42 (4.9%) | 0.83 |
| 10 | PC | 14 (1.4%) | 15 (1.7%) | 0.58 |
| 11 | B-IC | 9 (0.9%) | 11 (1.3%) | 0.50 |

The Fisher exact test (two-tailed) was used to compare groups.
*indicates statistically significant differences.

identified AQP2-negative tubules in which most cells expressed SLC8A1, potentially consistent with the aforementioned distal convoluted tubule *Slc8a1* expression in mice. Some A-IC subtypes have differential leading biological pathways following UPEC exposure and A-IC subtype A increased in relative frequency in response to UPEC. These findings indicate that A-IC subtype A may represent an innate immune variant. The role of *SLC8A1* in IC differentiation warrants investigation in future studies.

We performed scRNA-seq on 1861 human kidney cells of which 1066 were ICs. To contextualize murine experimental models to human pathophysiology, it will be important to compare human and murine ICs. Chen and colleagues enriched for murine ICs with c-KIT and performed scRNA-seq[22]. They classified 74 cells as PCs, 87 as A-ICs, and 23 cells as B-ICs[22]. Ransick et al.[30] performed scRNA-seq on 688 ICs. They did not enrich for

ICs but rather divided the kidney into the cortex, inner medulla, and outer medulla to evaluate zonal differences[30]. We demonstrated that murine-collecting duct cell-type marker expression appeared to be largely conserved in human collecting duct cells. Whether mice have comparable A-IC subtypes to humans will likely require a targeted single-cell evaluation of a larger number of murine ICs.

We identified scRNA-seq IC expression of innate immune proteins such as the immune and inflammatory mediator galectin 3 (*LGALS3*) and the *ADM* that we have previously reported in rodent ICs[15,40–42]. However, some key innate immune proteins previously reported in ICs, such as Lipocalin 2 (*LCN2/NGAL*), *RNASE7,* and toll-like receptor 4 (*TLR4*) had minimal expression or were not seen in human ICs on the single-cell level in our analysis[6,9]. The paucity of *LCN2/NGAL* and *TLR4* expression in

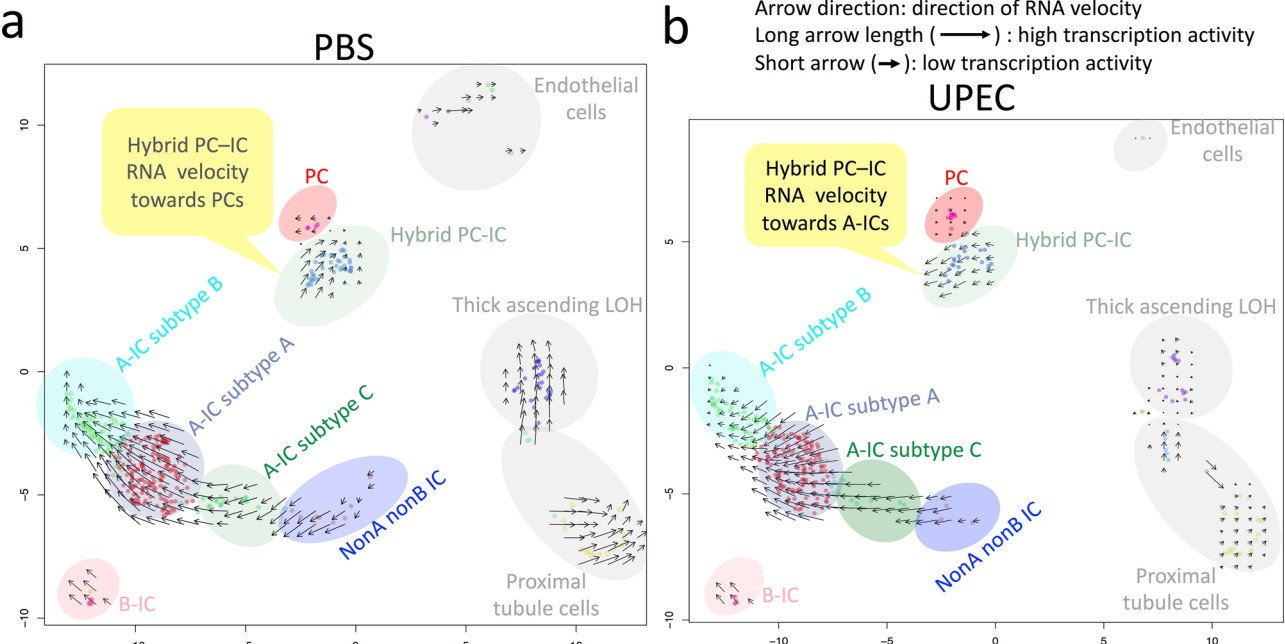

**Fig. 5 UPEC exposure results in human hybrid PC–IC RNA velocity toward A-ICs.** At baseline (**a**) represented by 1 h of control saline exposure, human hybrid PC–ICs have an RNA velocity that moves counterclockwise toward PCs. Following 1 h of UPEC exposure, hybrid PC–IC's RNA velocity changes directions toward A-ICs. Also, non-IC cell types, including proximal tubule, PCs, endothelial, and thick ascending loop of Henle (LOH) have much smaller arrows with UPEC exposure (**b**) compared to saline (**a**) indicating that these cells have decreased transcriptional activity in response to infection. In contrast, ICs had similar sized arrows with saline and UPEC exposure, indicating that they maintain their transcriptional activity when infected. The color shade of the collecting duct cell clusters matches the cluster color in Fig. 1, while the non-collecting duct kidney cell clusters are shaded gray.

ICs is consistent to what has been reported in murine ICs by our research group along with Ransick et al.[30] [https://cello.shinyapps.io/kidneycellexplorer/][15]. Chen et al.[22] did report some *TLR4* and *NGAL* mRNA expression in ICs, but several fold less than in PCs. We did evaluate pooled human ICs for *RNASE7* expression and identified it in ICs from 1 out of 4 kidneys (Supplementary Fig. 18), indicating that its expression may be intermittent depending on region, time point, and physiological conditions. *CISH* expression, which is induced in nonA nonB ICs in response to UPEC, has been demonstrated to regulate the innate immune response to *Mycobacterium tuberculosis* in the lung and spleen[43]. ICs phagocytosed bacteria over several minutes, and our cells for scRNA-seq were exposed to UPEC for a relatively short 1 hour time point. Future studies are needed to determine if other IC pathways, such as AMP expression or regulation of cell death, become more prominent in later time points following UPEC exposure.

Phagocytosis involves the cellular uptake of particulates >0.5 μm by the plasma membrane envelope[44]. "Professional phagocytes", myeloid-derived immune cells, including macrophages, neutrophils, and osteoclasts differentiate invading microbes from the microbiota and healthy cells, engulf the target into a phagosome, generate reactive oxygen species (ROS), secrete AMPs, and present antigens to other cells[45–49]. Of importance, robust acidification by V-ATPase of phagolysosomes creating an acidic microenvironment sufficient for killing most microbes is a hallmark characteristic of professional phagocytes[48,50]. For example, inhibiting V-ATPase with bafilomycin A1 suppresses macrophage bactericidal activity[51]. Certain epithelial cells phagocytose microbes but are less efficient at bacterial killing than professional phagocytes and have been described as nonprofessional phagocytes[52]. ICs represent an epithelial lineage that has

phagocytosis and antigen-presentation capabilities (Figs. 6–9) along with robust AMP expression and generous mitochondria to generate ROS; thus, ICs appear to have more comparable bacterial-killing potential to macrophages than the aforementioned nonprofessional phagocytes[8,53–55]. A-ICs have previously been demonstrated to be capable of high rates of apical endocytosis of dextran into cytoplasmic vesicles[56]. We speculate that IC phagocytosis of UPEC is an extension of characteristics shared with macrophages, including V-ATPase expression, redox potential, AMP secretion, and endocytosis capabilities[8,16,49,57,58]. Whether ICs can phagocytose cellular debris and bacteria other than UPEC in a similar manner to professional phagocytes remains to be determined.

Confocal imaging of a single-kidney tubule perfused in vivo with bacteria expands on prior in vitro techniques like cell culture and perfusion of dissected tubules. Here, we developed methodology to directly test in live mice whether phagocytosis is an IC function. Intravital microscopy allows for intricate study of dynamic cellular processes within functioning murine kidneys[59]. Past strategies to utilize intravital microscopy have involved systemic injection of intravital dyes or markers into mice to study proximal tubule endocytosis, kidney blood flow dynamics, and vascular or glomerular permeability[60]. When intravital microscopy is combined with sophisticated surgical techniques, difficult-to-evaluate early time points of infective processes can be studied[61]. Previously, bacteria were microperfused in proximal tubules of rat to evaluate blood flow, neutrophil recruitment, and urinary obstruction[62,63]. The vast majority of UTIs are due to ascending pathogens and would initially encounter the collecting duct[64,65]. Thus, we propose that the collecting duct is a critical tubular region to evaluate host-pathogen interactions. Our intravital model system effectively recreates a classical

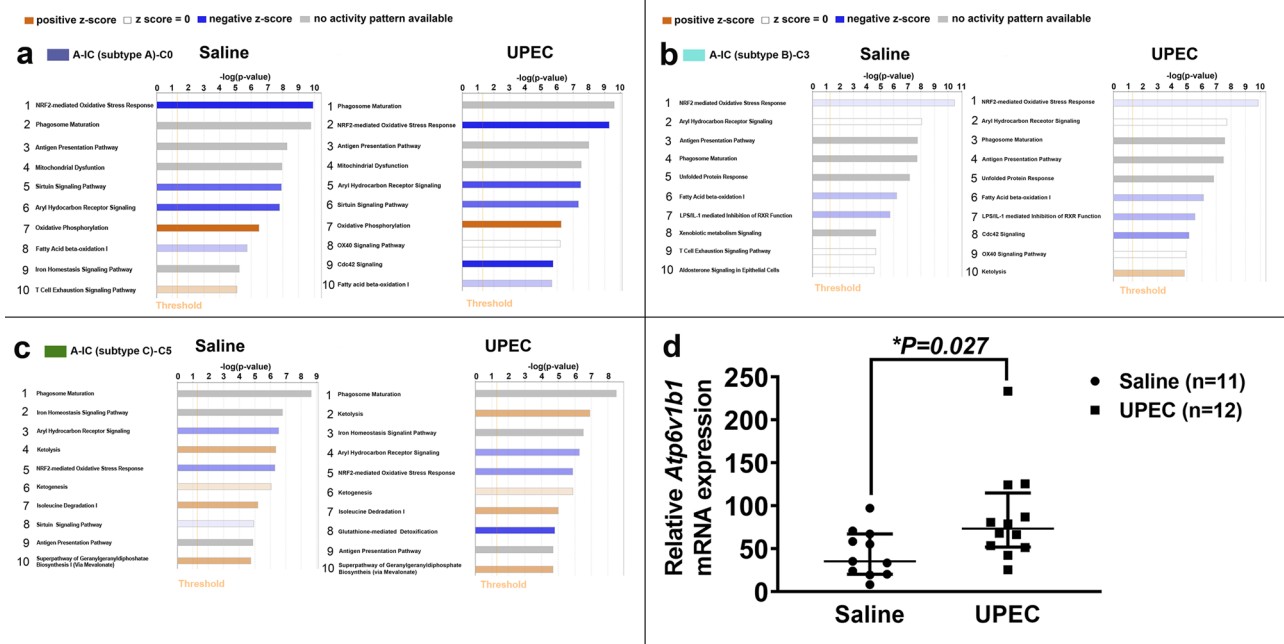

**Fig. 6 Human A-ICs, but not other kidney subtypes, have differential leading biological pathways following UPEC exposure and phagosome maturation is a leading A-IC subtype biological pathway.** The ten leading biological pathways ranked by p-value are presented and the colored box beside the cell type corresponds to the cluster color in Fig. 1. **a** With saline exposure, cluster 0/A-IC subtype A, nuclear factor, erythroid 2-like 2 (NRF2)-mediated oxidative stress response, phagosome maturation, and antigen presentation were the 3 leading biological pathways. With UPEC exposure, cluster 0/A-IC subtype A, phagosome maturation surpassed NRF-mediated oxidative stress response to become the leading pathway. TNF receptor superfamily member 4 (OX40) signaling and cell division cycle 42 (Cdc42) signaling appeared in the top ten pathways. **b** Cluster 3/A-IC subtype B also had differential rankings of the ten most involved pathways. Phagosome maturation moved from the fourth to third most involved pathway and Cdc42 signaling, OX40 signaling, and ketogenesis replaced xenobiotic metabolism, T-cell exhaustion signaling, and aldosterone signaling in epithelial cells as the eighth, ninth, and tenth most involved pathways with UPEC compared to saline exposure. **c** Cluster 5/A-IC subtype C was the remaining cell type that had differential top ten involved pathways between saline and UPEC exposure. Ketolysis moved from fourth to second, aryl hydrocarbon response went from glutathione-mediated detoxification appearing with UPEC-replacing sirtuin signaling pathway that moved off of the list. **a–c** Data were analyzed using IPA (QIAGEN Inc., https://www.qiagenbioinformatics.com/products/ingenuity-pathway-analysis). IPA reports a p value using a right-tailed Fisher's exact test. **d** "IC reporter" mice were challenged with UPEC strain CFT073 or saline transurethrally for 1 h. ICs were flow-sorted and Atp6v1b1 mRNA expression was analyzed. Atp6v1b1 mRNA expression increased by a median average of ~2-fold (35.23 (20.40–67.17) saline vs. 73.40 (51.97–115.00)) (UPEC) in ICs following transurethral inoculation of UPEC strain CFT073 versus saline; $n = 11$ mice in saline and $n = 12$ mice in UPEC group. Statistical significance (asterisk) between saline and UPEC exposure was determined by the Mann–Whitney test because the data were not parametric. The results are presented as the median and interquartile range. Source data are provided as a Source Data File.

single collecting tubule perfusion study, but in a live mouse with an intact kidney, blood, lymphatic, and neuronal supply along with retained ability to interact with immune cells.

Pyelonephritis pathophysiology has male versus female distinctions. For example, in humans, females are five times more likely to develop acute pyelonephritis but male mice have androgen-mediated increased severity of pyelonephritis[66,67]. Ransick and colleagues demonstrated differential male and female differences in proximal tubule cells, particularly in organic anion transporters[30]. Male ICs were evaluated in both our single-cell sequencing and intravital studies. Because of the role of ICs in the bacterial defense of the kidney and electrolyte balance, sex-related diversity in IC functions, such as phagocytosis and single-cell gene expression, will be key areas for future studies.

This study does have limitations. The UPEC exposure to human ICs in our scRNA-seq was in vitro to a single kidney and may have distinctions compared to in vivo UPEC exposure. In addition, our findings represented the anatomical region of the human kidney sampled and future studies with different locations may have regionalized diversity as demonstrated by Ransick et al.[30]. ICs comprised 57% of the cells in our scRNA-seq, enriched from the ~7% IC composition in kidneys at baseline[33]. Our sample did contain some other epithelial and endothelial cell

types, but was negative for CD45+ immune cells that were sorted off prior to enrichment of ICs (Fig. 1, Supplementary Fig. 1). With scRNA-seq, the target ICs could be identified and evaluated separately. Hematopoietic stem cells express c-KIT, but they do not likely represent the kidney resident myeloid cell populations[68]. We did not identify other c-KIT-expressing cells present in healthy human kidneys, such as stromal cells/telocytes[69]. Some isolated loop of Henle and proximal tubule c-KIT positivity has been previously reported on kidney immunohistochemistry consistent with our scRNA-seq results[70].

Murine intravital imaging limitations were also present due to the challenging nature of the single- tubule microperfusion experiments. Monitoring of a microinjection procedure under a fluorescence confocal/two-photon microscope brings advantages, but also challenges. First, a unique "angled" organ positioning required for a successful injection eliminated usage of a preferred two-photon scanning mode. In most cases, only a narrow band of a focal volume could be visualized, as dictated by the depth discrimination nature of two-photon microscopy. Switching to a confocal mode with a slightly open pinhole eliminated this problem. The second limitation we encountered was dictated by a motion artifact coming from a breathing of an anesthetized animal. Although this artifact can be easily eliminated while

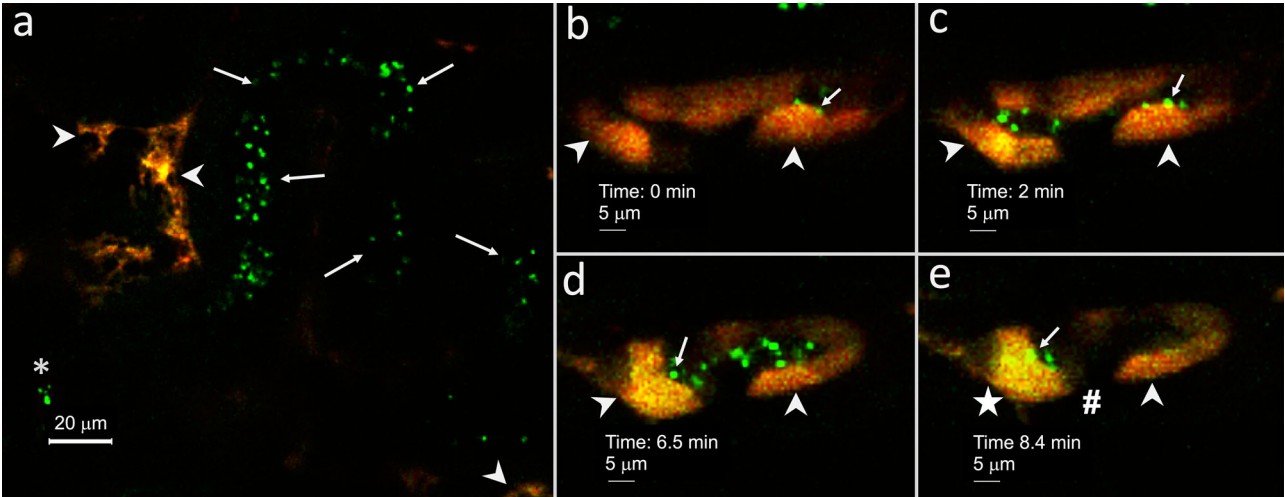

**Fig. 7 Intravital confocal microscopy demonstrates murine IC phagocytosis of UPEC in real time.** A single renal connecting and/or collecting tubule of a "IC reporter" mouse was perfused with GFP expressing UPEC and imaged during confocal intravital microscopy. **a** An example of the imaging with tdT expressing ICs (red) (arrows), the needle tip (asterisk) and GFP expressing UPEC (green) (arrows) flowing through a tubular lumen are presented. Approximately, 1 min after the UPEC injection into the tubule, target tdT+ ICs were identified and imaging was started with the 0 (**b**), 2 (**c**), 6.5 (**d**), and 8.4 (**e**) min time points presented. **b**, **c** UPEC (arrows) can be seen adhering to the lumen of IC (arrowheads) at early time points (**d**). At the 6.5 min time point, ICs have luminal UPEC aggregates (arrow) and punctate areas of yellow (arrowheads) consistent with internalization of the green UPEC into the red ICs. **e** At the 8.4 min time point, more UPEC uptake takes place in some ICs (star) compared to other ICs (arrowhead), while tubule areas with no tdT expression (pound sign) (presumed principal cells) have little-to-no green UPEC uptake. UPEC aggregates remain around the luminal IC surface (arrow). **a–e** Magnification 20×, with digital zoom. Each image is from the same tubule and imaging plane, but at a different time point. Because of breathing-induced cell movement, cellular morphology and position varied from image to image. Images are representative of intravital experiments on two individual "IC reporter" mice. Source data are provided as a Source Data File.

imaging on an inverted microscope, this particular experiment can be performed only on an upright microscope stand and with a dipping lens. Therefore, as much as we would like to collect XYZT series of acquisitions, we found it impractical in some of the experiments. A single-plane, XYT series were collected in such cases but resulted in a different number of images captured per time period between images. The magnification during intravital microscopy was 20× and *E. coli* are only 1–2 microns in length; therefore, UPEC aggregates rather than individual UPEC were visualized[71]. Microperfusion of a single collecting or connecting tubule was technically challenging and the flow of bacteria or bioparticles was not able to be predicted beforehand. Thus, target tubules would need to be found downstream and likely had some uptake of bacteria or bioparticles prior to the initial image. Future directions using different bacterial strains and growth conditions will lead to an increased understanding of bacteria-IC interactions. For example, the pyelonephritis strain that we used (UPEC CFT073) could be compared to a cystitis strain such as UPEC strain UTI-89 or the *E. coli* K12 strain is generally used to coat bioparticles[72]. Additionally, different bacterial growth conditions could be compared, for example, UTI-89 has been demonstrated to have increased type 1 pili formation if grown statically[73].

In conclusion, the findings of this study have expanded our knowledge of IC differentiation capabilities beyond polarity changes with acidosis. We have also demonstrated that ICs phagocytize UPEC and then acidify the phagolysosomes, thereby representing an epithelial cell that has convergently developed capabilities similar to myeloid-derived macrophages. The ability of ICs to differentiate into a range of subtypes and phagocytose UPEC raises the possibility of future treatment strategies based on shifting IC differentiation toward innate immune defense subtype (s), and enhancement of these cells' processes such as phagocytosis and acidification to either complement or replace antibiotics in clearing an ascending UTI.

## Methods

**Regulatory approval**. Murine studies were approved by the Institutional Animal Care and Use Committee (IACUC) at the Indiana University School of Medicine, protocol number 11333, and adhered to the "NIH Guide for the Care and Use of Laboratory Animals". Human tissue was obtained from the Cooperative Human Tissue Network (CHTN) Midwest Division (Columbus, OH) and details regarding the policies and procedures that govern collection of specimens and distribution to investigators are found at their website: [https://www.chtn.org/policies/index.html]. The key polices include protection of subjects with all policies being consistent with repository guidance and regulations from the Office of Human Research Protections in the Department of Health and Human Services (OHRP, DHHS), review/approval of the local Institutional Review Boards (IRB), specimen derivation from residual sample material removed as a component of routine medical care in accordance with state and local laws, each CHTN institution has received human subject assurance from the OHRP that the institutions comply with the "Common Rule" (45 CFR part 46), each CHTN division has local IRB approval to collect and distribute biospecimens that are reviewed yearly, donor identities and identifying information is not provided to investigators, and participating investigators must document IRB review (and approval if deemed necessary) by the IRB where the proposed research will be taking place. The use of CHTN samples for this study was reviewed by the Indiana University Institutional Review Board and was exempt (protocol 1802253259) because patient samples did not include identifying information and permissions and consents for human sample collection were from the CHTN and its member institutions. The research complied with all relevant ethical regulations for work with human participants.

**Human kidney tissue**. Fresh human kidney biopsy samples were obtained from the CHTN[74]. The kidney tissue consisted of normal kidney margins from kidney resection surgeries. The tissue was cut into small, ~2–4-mm pieces, placed in sterile DMEM, and shipped overnight in cold packs. Supplementary Table 1 outlines the clinical characteristics from the patients that the kidney margins were obtained from. For immunofluorescence, microscopy samples were obtained from the CHTN[8,74,75].

**Mice**. *Strains and generation of "IC reporter" mice*. V-ATPaseB1-cre transgenic mice (kindly provided by Dr. Raoul Nelson, University of Utah, UT) in which 7-kb B1 promoter drives the cre expression in ICs were used[76]. The V-ATPaseB1-cre mice were crossed with tdTomato-loxp homozygous mice (B6.Cg-Gt(ROSA) 26sor^tm9(CAG-tdTomato)) (Stock no. 007909, Jackson lab, Maine). The progeny of the V-ATPase-B1 and tdTomato-*loxp* mice were genotyped for the cre gene (see Supplementary Table 5). Mice positive for the cre gene were used and termed "IC reporter" mice[16]. All mice were on a C57BL/6 background.

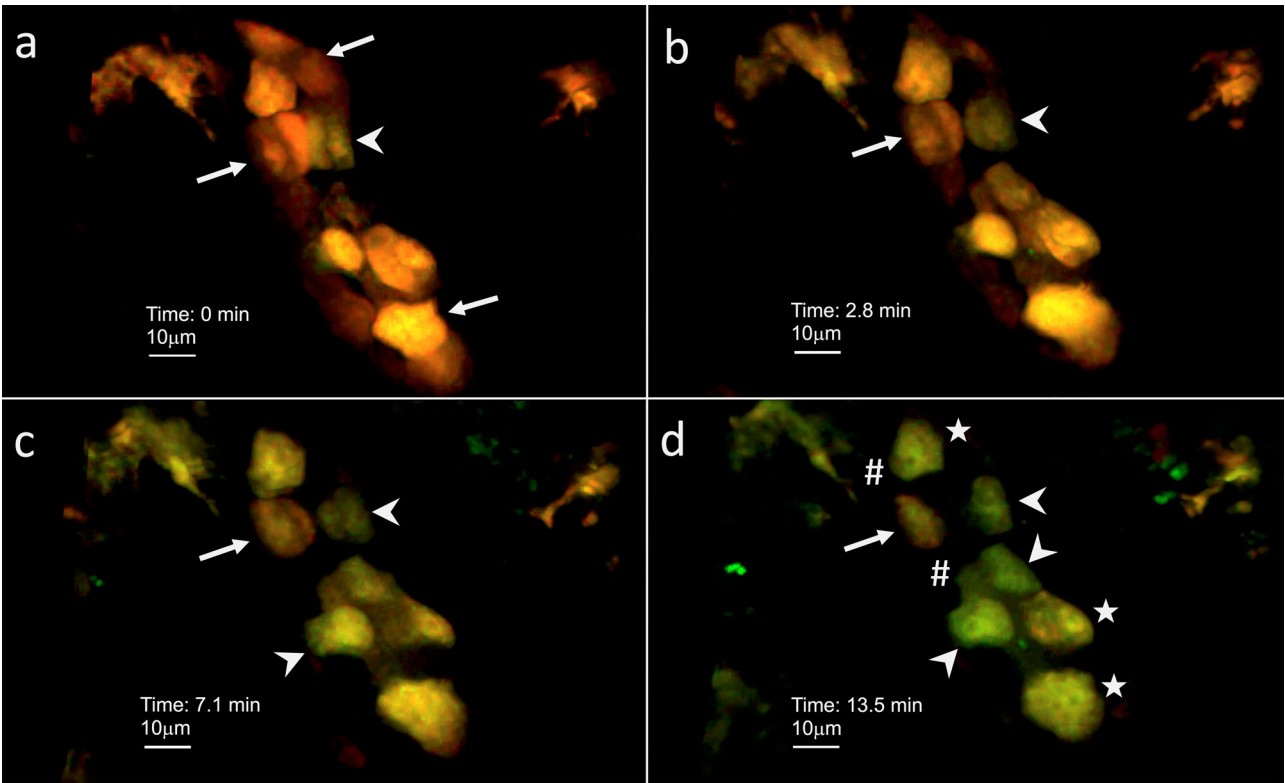

**Fig. 8 Murine ICs internalize and acidify *E. coli*.** An "IC reporter" kidney tubule with tdT-expressing ICs (red) was perfused with pHrodo Green *E. coli*-coated BioParticles for phagocytosis that fluoresce green when acidified. **a** At the 0 min time point red ICs are seen (arrow). Because the 0 min time point started ~1 min after injection when we identified our target tubule and narrowed the focus, some cells may be already starting to uptake bioparticles (arrowhead). **b** Increasing green fluorescence occurs in patchy areas in the cytoplasm in an isolated IC (arrow) as the experiment progresses to the 2.8 min time point but not some remain red (arrow). **c** At the 7.1 min time point, most ICs are increasingly green (arrowheads), but an isolated IC remains red (arrow). **d** At the 13.5 min time point, variable uptake/acidification can be seen with relatively high (arrowhead), medium (star), or low (arrow) green fluorescence visualized. No appreciable green fluorescence is present in red fluorescence-negative cells (presumed PCs) (pound symbol) within the tubule. **a–d** Magnification 20×, with digital zoom. Images representative from imaging of three tubules from two mice. Each image is from the same tubule and imaging plane, but at a different time point. Because of breathing-induced cell movement, cellular morphology and position will vary from image to image. Images are representative of intravital experiments on two individual "IC reporter" mice. Source data are provided as a Source Data File.

*Housing and husbandry*. All mice used for this experiment were bred and kept in a pathogen-free housing facility at the Indiana University School of Medicine, Indianapolis, IN, which is Association for Assessment and Accreditation of Laboratory Animal Care International (AALAC) accredited. The day light cycle of the colony is 7 AM–7 PM and mice are kept at an ambient temperature of 65–75 °F with 40–60% humidity. Mice for experiments were all male, except those used for in vivo UTI experiments, and kept separated by sex, with no more than five male mice being in a single cage at one time. Bedding material used was wood shavings for the main base with tissue paper also being provided to mice for bedding. Mice for these breeding strains were bred with 1 male with 2 females under normal, nonaltered conditions with water, light, or food. Animals were weaned from breeding pairs at 28 days and separated into separate cages depending on date of birth as well as sex. Animals were cared for by vivarium staff before being selected for the experimental procedures.

**Bacteria**. GFP expressing uropathogenic *E. coli* (UPEC) strain CFT073 (originally a gift from Matthew A. Mulvey, University of Utah) was used in the study. It was stored at −80 °C in 50% glycerol until use. For murine experimental UTI induction and intravital tubular perfusion, the following bacterial growth protocol was used: (a) a culture tube was prepared with 4 ml of Luria broth and labeled time and date, (b) CFT073 GFP expressing UPEC was retrieved from −80 °C, (c) the pipette tip was used to scrape some frozen bacterial stock and the tip was placed in the tube to transfer over frozen bacteria, (d) the tube with Luria broth and bacteria was placed into a 37°C incubator overnight for ~16–18 h, and (e) the next day, the bacterial stock was spun down and resuspended in sterile PBS at a concentration of $1 \times 10^8$ CFT073 in 50 μl of sterile PBS for use in experimental UTI.

**Enrichment of human kidney ICs with MACS**. Upon arrival, kidney biopsy tissues were further dissected into small pieces and transferred to a C tube (Miltenyi Biotec, Bergisch Gladbach, Germany) containing 5 ml Liberase TL (500 μg/ml) (Sigma, St Louis, MO) with DNase I (100 μg/ml) (Sigma, St. Louis, MO) containing

DMEM media. Single-cell suspension was prepared with GentleMACS (Miltenyi Biotec) using lung 02_01 program. Cells were incubated at 37 °C for 15 min and then program spleen 04_01 was run, cells were again incubated for 15 min at 37 °C. Cell suspension was filtered with 70-μm basket filter (Fisher Scientific, Waltham, MA). Cells were centrifuged for 10 min at $1000 \times g$. RBCs were lysed with 1X RBC lysing buffer (Biolegend, San Diego, CA) for 5 min on ice with agitation, cells were then washed with DMEM and resuspended in 5 ml fresh DMEM and filtered with 70-μm basket filter. Dead cells were removed from cell suspension using dead cell removal microbeads (150 μl beads/$10 \times 10^6$ cells, Cat No. 130-090-101, Miltenyi Biotec) on LS column. Live cells were counted and CD45+ immune cells were removed using anti-human CD45 microbeads (20 μl beads/$10 \times 10^6$ cells, Cat. No. 130-045-801, Miltenyi Biotec) before the ICs were enriched. CD45-depleted cells were centrifuged at $300 \times g$ for 10 min and counted and then incubated with 100 μl FcR blocking reagent and 100 μl anti-human CD117 (c-Kit) microbeads (Cat no. 130-091-332, Miltenyi Biotec) at 4 °C. Cells were applied over MS column and c-Kit+ (CD117+) cells were flushed out of the column with 1 ml MACS buffer (PBS containing 0.5% bovine serum albumin (BSA) and 2 mM EDTA). RNA quality was tested on Agilent Bioanalyzer (Agilent, Santa Clara, CA). Enriched cells were tested for mRNA expression of *ATP6V1B1, SCL4A1, and SLC26A4* gene by RT-PCR to confirm the relative IC enrichment (Supplementary Fig. 2).

Exposure of dissociated kidney cells to UPEC and saline: UPEC was grown overnight in Luria broth on an orbital shaker at 37 °C. An aliquot of bacterial broth was pelleted and suspended in sterile PBS. Optical density of the culture was measured at 600 nm. Based on this calculation, OD600 of $1 = 8 \times 10^8$ cells/ml, $1 \times 10^5$ bacterial cells were estimated (in ~20 μl bacterial suspension). Enriched IC cells as prepared above were equally divided in 2 wells of a 96-well U-bottom plate (Fisher Scientific) and incubated for 1 h at 37 °C and 5% $CO_2$ in 180 μl DMEM containing 10% FBS with $1 \times 10^5$ UPEC cells or equal volume of sterile PBS alone. After incubation, cells were centrifuged at $300 \times g$ for 10 min and media was carefully removed, cells were washed again with PBS, suspended in 50 μl PBS (without Ca2+ or Mg2+), and were taken to the IUSM genomics core for 10× chromium single-cell preparation and sequencing.

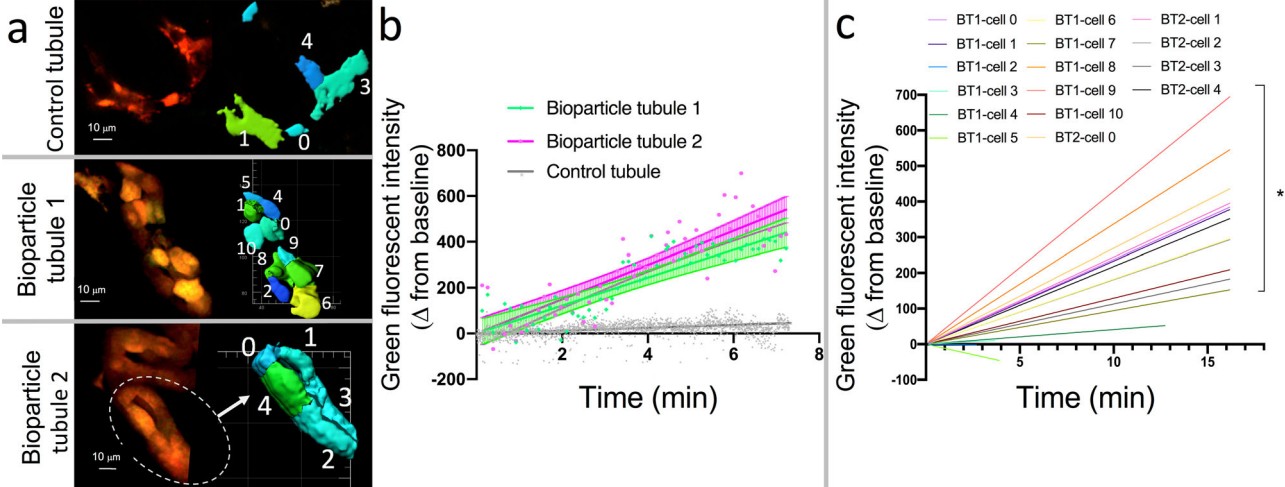

**Fig. 9 Imaris quantification of green fluorescence following tubular microperfusion with pHrodo Green *E. coli* BioParticles demonstrates increased uptake and acidification by ICs over time. a** Tubules of "IC reporter" mice with red ICs along with Imaris segmentation profiles are presented for control (top panel) and pHrodo Green *E. coli* BioParticle perfused tubules (middle and bottom panels). Segmentation cartoons (right panels) are not to scale. (**b**) Because pHrodo Green BioParticles fluoresce green when internalized and acidified in phagolysosomes, we compared the mean change in green fluorescent intensity for the IC component of each tubule over time. The change in green fluorescence from baseline over time increased significantly for both coated *E. coli*-exposed bioparticle tubule 1 (BT1) and bioparticle tubule 2 (BT2) compared to the control tubule (CT). Linear regression was used to compare tubules and cells, and the results are presented as the mean ± 95% confidence interval. For BT1, BT2, and CT, respectively, the $R^2$ was 0.28, 0.12, and 0.04, the slope was 60.9, 74.8, and 6.8, the DFn was 1467, 1264, and 17264, the $F$ was 66.5, 104.6, and 217.5, and the Sy.x was 329.8, 244.8, and 56.5. The slopes of BT1 and BT2 were both significantly higher than the slope of CT ($P < 10E-15$), but not different from each other ($P = 0.221$). **c** Within the bioparticle-perfused tubules, 12 of 16 tdT+ ICs had a slope that increased significantly compared to a zero-slope line. Thus, there is some heterogeneity in IC phagocytic capabilities. During linear regression, Graphpad Prism compares slopes by calculating a two-tailed p value to test the null hypothesis that the slopes are identical. For the tubule analysis (**b**), slopes were compared to one other tubule (BT1 vs. BT2, BT1 vs. CT, and BT2 vs. CT). For the cell analysis (**c**), the slope of each cell was compared individually to a zero-slope line. Multiple comparisons were not used. Analysis included 2 bioparticle tubules (one with 11 ICs and the other with 5 ICs) and 1 control tubule (with 5 ICs) from one mouse each. Source data are provided as a Source Data File.

**Single-cell RNA sequencing (ScRNA-seq).** The single-cell 3′ RNA-seq experiment was conducted using the Chromium single-cell system (10× Genomics, Pleasanton, CA) and the NovaSeq 6000 sequencer (Illumina, CA). Each single-cell suspension was first counted on the countess II FL (Thermo Fisher Scientific) as well as under light microscope for cell number, cell viability, and cell size. Single-cell preparation included resuspension, centrifugation, and filtration to remove cell debris, dead cells, and cell aggregates. The resulting single-cell suspensions had viabilities of 75% and 83% with concentrations of 200 and 510 live cells per μl of sample for saline- and UPEC-exposed cells, respectively. Five- to ten- thousand cells per sample were added to a single-cell master mix, following the Chromium Single Cell 3′ Reagent Kits v2 (User Guide, CG00052 Ver B) along with the single-cell gel beads and partitioning oil in separate wells of a Single Cell A Chip, the single-cell master mixture containing an aliquot of single-cell suspension was loaded to the Chromium controller for gel bead-in emulsions (GEM) generation and barcoding, followed by cDNA synthesis and library preparation. At each step, the quality of cDNA and library was examined by Bioanalyzer. The resulting libraries were sequenced using a custom program for generating 26 and 98-bp paired-end reads on an Illumina NovaSeq 6000 (Illumina NovaSeq 6000 control software version 1.6.0 and the Real time analysis version 3.4.4).

**ScRNA-seq data analysis.** *Data processing and filtering.* CellRanger version 3.0.2 (10× Genomics) was utilized to process the raw sequence data generated[77]. Briefly, CellRanger uses bcl2fastq to demultiplex raw base sequence calls generated from the sequencer into sample-specific FASTQ files. The FASTQ files were then aligned to the reference genome with RNA-seq aligner STAR. The aligned reads were traced back to the individual cells and the gene expression level of individual genes was quantified based on the number of UMIs (unique molecular indices) detected in each cell. The filtered gene-cell barcode matrices generated with CellRanger were used for further analysis with the R package Seurat development version 3.0.0.9000[78,79]. QC of the data was implemented as the first step in our analysis. We first filtered out genes that detected in less than five cells and cells with less than 200 genes. To further exclude low-quality cells in downstream analysis, we used the function Outlier from R package scatter together with visual inspection of the distributions of the number of genes, UMIs, and mitochondrial gene content[80]. Cells with high or low number of detected genes/UMIs were excluded. In addition, cells with greater than 25% mitochondrial reads were also filtered out. After removing likely multiplets and low-quality cells, the gene expression levels for each cell were normalized with the NormalizeData function in Seurat. To reduce variations sourced from different number of UMIs and mitochondrial gene

expression, we used the ScaleData function to linearly regress out these variations. Highly variable genes were identified with the FindVariableFeatures function (selection.method = "vst").

*Cluster analysis.* To integrate the single-cell data from the treated and untreated samples, functions "FindIntegrationAnchors" and "IntegrateData" from Seurat v3 were implemented with a dimensionality of 30. The integrated data were then scaled and PCA performed. Clusters were identified with the Seurat functions "FindNeighbors" and "FindClusters" using a resolution of 0.8 and the first 12 PCs[78]. The "FindConservedMarkers" function was used to identify cell cluster marker genes. To compare average gene expression within the same cluster between cells of different samples, AverageExpression function was applied. "FindMarkers" function was also used to investigate the differences of gene expression induced by treatment within the same cell cluster. The cell clusters were visualized using the t-Distributed Stochastic Neighbor Embedding (t-SNE) plots and Uniform Manifold Approximation and Projection UMAP) plots. R packages ggplot2 [https://ggplot2.tidyverse.org] and ggrepel [https://github.com/slowkow/ggrepel] were used to plot the average gene expression[81,82]. Violin plots (VlnPlot) and feature plots (FeaturePlot) were used to visualize specific gene expressions across clusters and different sample conditions. Figures of canonical pathways were made with the Ingenuity Pathway Analysis Software. The raw datasets for scRNA-seq have been deposited in the National Center for Biotechnology Information (NCBI) Gene Expression Omnibus (GEO) repository under accession number GSE159805.

*Pathway analysis.* Canonical pathway analysis was performed using Ingenuity Pathway Analysis [QIAGEhttps://www.qiagenbioinformatics.com/products/ingenuity-pathway-analysis] (Qiagen, Hilden, Germany) to identify biological pathways associated with single-cell clusters (April 2019 release, analysis performed in August, 2019)[83]. Entrez gene ID, log2 fold change, and adjusted p-values from Seurat scRNA-seq are the IPA inputs. The log2 fold change is the expression relative to other clusters. Seurat uses the Wilcoxon rank sum test to generate a p-value for gene expression and a Bonferroni correction to account for a false discovery rate and to calculate an adjusted p-value. Ingenuity reports a p-value regarding pathway enrichment versus a random list of genes within a cluster using a right-tailed Fisher's exact test to calculate the probability that assigned biological functions are not due to chance alone[84].

**RNA velocity analysis.** Mapping position sorted.bam files generated by cellRanger were converted to cellsorted.bam files through the velocyto pipeline v0.17 (velocyto.org) to obtain.loom files for each experimental condition. These.loom files

along with their associated UMAP positions and principal component tables extracted from the merged Seurat file were then fed individually into the RNA Velocity pipeline as described in the Velocyto.R Dentate Gyrus/loom tutorial[32]. The default settings described in the tutorial were used, except for tSNE positions that were overwritten with the associated UMAP positions from the merged Seurat object, as well as the principal component table. This generated an RNA velocity figure mapped using the merged Seurat object cell positions for both conditions. The code used for the RNA velocity is presented in Supplementary Software 1.

**Intravital imaging of kidney**. *Mouse conditions*. Eight experimental intravital sessions and 1 control session (bleaching control) were undertaken. Animal groups used were all male and selected randomly among the youngest available litter of V-ATPase-cre$^+$tdT$^+$ ("IC reporter") mice. The experimental unit in these procedures was a single animal; no more than a single mouse was used per intravital session.

*Surgical preparation*. All experiments were performed in the afternoon between the hours of 1:00–4:00 PM Eastern Standard Time and took place in the microscopy core at Indiana University with surgical exposure of the kidney occurring in an approved animal procedure facility adjacent to the room housing the microscope. After exposure of the kidney, the mouse was carefully transported to the microscope stage equipped with warming pads to maintain the animal's body temperature. Live animal imaging was used to visualize the pHrodo Green *E. coli* BioParticles and uropathogenic *E. coli* (UPEC strain CFT073) phagocytosis by tdTomato expressing ICs (V-ATPase-cre$^+$tdT$^+$, "IC reporter" mice) in real time. To prepare the animal for intravital imaging, the animal was placed in an induction box providing 3–5% isoflurane, and then transferred to 2–2.5% isoflurane via nose cone. The animal was placed in right lateral decumbency on a 37 °C heating pad and hair was clipped from the left abdomen. The shaved region was cleaned with betadine and then wiped with 70% isopropyl alcohol using alternating rounds of application, allowing the area to dry after the last application. A ~1 cm vertical left flank incision was made, so that the left kidney could be externalized. A custom-modified plastic petri dish bottom with an approximately 0.5 cm opening was superglued to the surface of the left kidney. Saline was applied on the surface of the exposed organ to prevent drying. The animal was then positioned on a custom-made platform lined with a heating pad allowing minimal heat lost, to insure the petri dish/kidney stability by clamping. Next, the animal was transferred to a microscope stage of an upright confocal/2-photon microscope. Commercially obtained *E. coli* bioparticles (pHrodo Green *E. coli* BioParticles, Cat no. P35381, Molecular probes, Invitrogen, Waltham, MA) or GFP-UPEC strain CFT073 were injected into a collecting duct using a 5–10 micron diameter beveled glass micropipette (Clunbury Scientific, Bloomfield Hills, MI), a microelectrode holder (1.0 mm) from World Precision Instruments (Model MPH4), and New Era Syringe Pump (Model NE-300) for air-driven injection. GFP UPEC strain CFT073 was grown as described for the scRNA-seq experiment. The holder and pipette were stabilized and navigated with a Narishige Coarse Manipulator (Model MMN-1, Tokyo Japan). Animals were constantly monitored throughout the experiment. Due to the terminal nature of the procedure, no post-procedural monitoring was needed. Once imaging was complete, the animal was euthanized by overdose of isoflurane and cervical dislocation.

*Sample size and mouse characteristics*. Ten mice were used to obtain 2 successful tubule microperfusions for bacteria and 2 for bioparticles; additionally, one control mouse was used ($N = 11$ total male mice were used). Experiments were unsuccessful because the tubular lumen was not able to be canulated resulting in injection into a blood vessel or the interstitium, the needle broke before bacteria/bioparticle injection, or the needle became repeatedly clogged with bacteria/bioparticles. The number of animals was not determined initially due to the pilot nature of this study. Results and data were obtained until the research team concluded that sufficient information was available. The experiments were performed on 11 separate occasions. The age of mice ranged from 55 to 103 days, median age: 55 days, mean age: 61 days; the mouse weight ranged from 19.3 to 22.4 g, median weight: 20.3 g, mean weight: 20.6 g.

*Imaging*. Microinjection experiment was conducted under a monitoring of a confocal microscopy using Leica SP8 confocal/2-photon system equipped with a Leica HC Fluotar L 25×/0.95 water dipping objective lens (Leica Microsystems, Wetzlar, Germany), available at the ICBM Imaging Facility, Indianapolis, IN. Images (12 bit, XYT, or XYZT series) were collected at a frame size of 512 × 512 pixels, 600 Hz scanning rate, zoom factor of 1–2, bidirectional X scanning mode, and a sequential illumination scanning mode set up for two sequences with two HyD detectors: green (488-nm excitation/500–550-nm emission), and orange (552 nm excitation/560–650 nm emission). For most experiments, a confocal pinhole was set to I.5–2.5 AU to visualize a bigger area of the kidney in one focal plane, as the organ needed to be positioned under an angle to facilitate a needle engagement and further injection. This unique positioning requirement eliminated usage of a two-photon scanning mode, since it would support visualization of a very restricted area of the kidney (a narrow band).

*Image analysis*. Quantitative measurements were performed using Imaris software (Bitplane, Concord, MA). Prior to analysis, the green channel was processed with a smoothing median filter (3 × 3 × 1). Segmentation of cells in the tubules was performed based on the red channel, using the "Surfaces" module. Segmented cells were tracked over the duration of the acquisition and intensity values for each channel were used to generate graphs. Change in green fluorescence

over time was determined by subtracting the green fluorescent intensity relative values from baseline defined as the *y* intercept. Differences between control and bioparticle-exposed cells were evaluated using linear regression (GraphPad Prism Software, San Diego, CA). Images of bioparticle-exposed cells were taken as a z-stack with images from the focal plane that consistently had the optimal images. The control images were taken from a single focal plane.

**Murine in vivo UTI and isolation of ICs**. *In vivo murine UTI*. UPEC strain CFT073 was grown overnight in an orbital shaker at 37 °C for 18–24 h in the Luria broth. Optical density of the culture was measured at 600 nm. UTI was induced in "IC reporter" female mice by inoculating $1 \times 10^8$ CFT073 in 50 μl PBS transurethrally[13,16,85,86]. Briefly, female mice were anesthetized with isoflurane. A syringe, equipped with a soft polyethylene tubing with an outside diameter around the needle of 0.61 mm, was filled with the bacteria solution or saline and the tip covered in lubricating gel to ensure smooth insertion. The anesthetized mouse's bladder was palpated to determine if the bladder is full and if full, the urine was discharged using light pressure to ensure no harm to the mouse. The area was then cleaned with an alcohol swab and the urethra was exposed by manipulating the mouse. Upon exposure of the urethra, the lubricated syringe was carefully inserted in the urethral opening and the bacterial solution or saline was slowly injected to not damage the bladder.

*Atp6v1b1 mRNA expression following UTI*. One hour later, mice were euthanized and the kidneys were dissected out, single-cell suspension was prepared using Accumax enzymatic solution (Innovative Cell Technologies Inc., CA) and gentleMACS disssociator (Miltenyi Biotec). Cells were surface-stained with anti-mouse APC-conjugated CD45 (1:250 dilution, Clone 30-F11, Cat no. 17-0451, eBioscience, San Diego, CA) for 30 min at 4 °C. Cells were washed and resuspended in PBS containing 2% BSA and viable CD45-tdTomato-positive (tdT+) ICs were flow-sorted using FACSAria Flow cytometer (Becton Dickinson, Franklin Lakes, KY) at the Indiana University School of Medicine flowcytometry core. *Atp6v1b1* mRNA expression in sorted ICs was measured by RT-PCR.

*Sample size*. With prior experiments evaluating the murine transcriptional response to experimental UTI, a broad standard deviation occured[53]. Therefore, we have developed the strategy of initially experimenting with 4–6 mice, depending on availability, per group (saline vs. UPEC inoculation). If significance is achieved or if there is no trend toward a difference the experiment is complete. If a trend toward significance is obtained, the experimental number is expanded to another 4–6 mice per group.

**Immunofluorescence and confocal microscopy**. Seven-μm-thick paraffin-embedded human kidney sections were stained with polyclonal rabbit anti-human c-KIT antibody (Cat no. A450229-2, Agilent Dako, Santa Clara, CA) to localize collecting duct ICs at 1:100 dilution overnight at 4 °C. To visualize, c-KIT, anti-Rabbit AF488 (1:600 final dilution) and to visualize V-ATPase, an antibody against the VATPASE E1 subunit (1:200 dilution) followed by anti-chicken Cy3 (1:600 final dilution) secondary antibody was used (Jackson Immunoresearch, West Grove, PA)[87]. For SLC8A1 immunolabeling, a 1:100 dilution of polyclonal rabbit anti-human SLC8A1 (Sigma Aldrich, St. Louis, MO) was used along with an anti-rabbit AF488 at 1:600 final dilution(Jackson Immunoresearch, West Grove, PA). To visualize V-ATPase, polyclonal chicken anti-human ATP6V1E1 (Sigma Aldrich, MO) at 1:200 dilution was used followed by anti-chicken Cy3 (1:600 final dilution). Polyclonal goat anti-human AQP2 (Santa Cruz Biotechnology, Dallas, TX) was used at a 1:200 dilution and visualized with 1:400 final secondary concentration of anti-goat Dylight405. For EGR1, HSP70 (HSPA1A), E1 immunolabeling (1:200 dilution), monoclonal Rabbit Anti-human EGR1 was stained at a 1:100 dilution followed by a secondary stain with a 1:600 final concentration of anti-rabbit AF488 (Jackson Immunoresearch, West Grove, PA). V-ATPase E1 staining was performed using a polyclonal chicken anti-human ATP6V1E1 antibody (Sigma Aldrich, St. Louis, MO) and visualized with an anti-chicken Dylight405 secondary at 1:200 final concentration. Last, monoclonal mouse anti-human HSP70 3A3 (Santa Cruz Biotechnology, Dallas, TX) was stained at 1:50 and visualized with an anti-mouse Cy3 secondary antibody at a final concentration of 1:600. Sections were imaged using a Keyence BZ9000 microscope (Keyence Corporation, Osaka, Japan) and/or Leica SP8 confocal. Validation images were obtained from the Human Protein Atlas version 19.3 [http://www.proteinatlas.org][29,88,89].

**RT-PCR of IC marker genes in enriched human cells**. Human kidney biopsy samples from four individual patients were processed. Single-cell suspension was prepared, dead cells were removed using dead cell removal microbeads, and c-KIT + and c-KIT− cells were magnetically enriched after depletion of CD45$^+$ immune cells as described earlier. The cells were pelleted, and RNA was prepared using RNeasy plus mini kit (Qiagen). cDNA was prepared using high-capacity cDNA reverse transcription kit (Applied Biosystems, Foster City, CA). In all, 1 μl cDNA was amplified with *ATP6V1B1*, *SLC4A1*, *SLC26A4*, and *RNASE7* primers (Kicqstart Primers, Sigma) in a 20 μl reaction on CFX connect real-time system (Bio-Rad Laboratories, Hercules, CA). *GAPDH* was used as internal control. Relative mRNA expression compared to *GAPDH* was calculated using ddCT method. Primer sequences for all PCR experiments are reported in Supplementary Table 5.

**In vitro *E. coli* phagocytosis assay**. To further confirm the live animal intravital imaging bacterial uptake, pHrodo *E. coli* BioParticle uptake was measured in vitro by flow cytometry. This method uses a unique pHrodo dye that fluoresces in response to an acidic environment. Kidney cell suspension was prepared from "IC reporter" mice using Accumax solution (Innovative Cell Technology, San Diego, CA) in combination with GentleMacs (Miltenyi Biotec). In this mouse strain, ICs are endogenously marked by tdTomato (tdT) red fluorescence. In total, 250,000 cells were surface-labeled with anti-mouse CD45 conjugated with APC (1:250 dilution, clone 30-F11, Cat no. 17-0451, eBioscience, San Diego, CA) for 30 min at 4 °C and then incubated with pHrodo Green-*E. coli* BioParticles or media alone and incubated for 15 min at 37 °C as recommended (Invitrogen, CA). After that cells were kept on ice to stop the reaction. Cells were immediately acquired on Attune Flow cytometer (Invitrogen, Carlsbad, CA) to measure green fluorescence on CD45$^-$tdT$^+$ (intercalated cell) and CD45$^+$ immune cells. Data were analyzed using FlowJo version 10 software (Becton Dickinson, Franklin Lakes, NJ). The gating strategies for flow cytometry experiments are presented in Supplementary Fig. 21.

**Statistical evaluation**. Unless stated otherwise, statistical evaluation was performed using GraphPad Prism. The Fisher exact test was used to compare percentages or proportions. Continuous variables were tested for normalcy with the Shapiro–Wilk test. If the data were parametric, a two-tailed *t* test was used, while the Mann–Whitney, also two-tailed, was used for nonparametric data. Linear regression was used to compare slopes.

**Figure generation**. Intravital figures were generated using Microsoft PowerPoint, GraphPad Prism, Adobe Photoshop, Leica Application Suite X, and/or Imaris software. Background fluorescence was only removed from nontargeted fluorescent cells/tubules on images with numbered cells presented as segmentation cartoons. Any brightness and contrast adjustments were performed uniformly across all images in a time series. Immunofluorescent microscopy and confocal microscopy figures were generated using Microsoft PowerPoint, Leica Application Suite X, and/ or Keyence BZ Analyzer software. Image adjustments included black balance along with brightness and contrast adjustments. Canonical pathway figures were generated using IPA software[83].

**Key resources**. Key resources for the experimental methodology are presented in Supplementary Table 6.

**Reporting summary**. Further information on research design is available in the Nature Research Reporting Summary linked to this article.

## Data availability
Contacting the originating investigators or source companies is needed to obtain key biological material. The scRNA-seq processed data can be found at https://hpcwebapps.cit. nih.gov/ESBL/Database/IU-Data/Human-c-Kit-Sorted-Single-Cell-RNASeq.htm. The scRNA-seq raw data are available at NCBI gene expression omnibus (GEO). Its GEO accession number is GSE159805. Kidney Cell Explorer[30] [https://cello.shinyapps.io/ kidneycellexplorer/] was used to compare our scRNA-seq findings to comparable past murine studies including GSE129798. For findings not related to the aforementioned links, we have submitted source data with this paper. Source images from the Human Protein Atlas version 20.0 may be obtained at https://www.proteinatlas.org/ENSG00000120738-EGR1/tissue/kidney#img, https://www.proteinatlas.org/ENSG00000204389-HSPA1A/ tissue/kidney#img, and https://www.proteinatlas.org/ENSG00000183023-SLC8A1/tissue/ kidney#img. Additionally, other data supporting this publication can be obtained by contacting the lead author. Source data are provided with this paper.

## Code availability
Code used for the RNA velocity analysis presented in Fig. 5 is available as Supplementary Software 1.

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

## Acknowledgements

The work was supported in part by the National Institute of Diabetes and Digestive and Kidney Diseases R01DK106286, Eli Lily Foundation (A.L.S. and D.S.H.), and Indiana Clinical and Translational Sciences Institute (CTSI) funded in part by award number UL1TR002529 from NIH (V.S.). We acknowledge Raoul Nelson of the University of Utah for his assistance with our mouse models.

## Author contributions

A.L.S., D.S.H., V.S., and T.H. supervised the study. A.L.S., D.S.H., V.S., T.H., M.H., A.Z., and M.M.K. developed the experimental protocol. V.S., S.A., M.M.K., A.Z., P.M., X.X., and H.G. performed the experiments. H.G., D.S.H., A.L.S., H.G., and M.M.K. performed the data analysis. A.L.S., V.S., D.S.H., S.A., T.H., X.X., H.G., M.H., and M.M.K. performed the critical edits.

## Competing interests

The authors declare no competing interests.
