## [Peer Review File · Nature Communications]

REVIEWER COMMENTS

Reviewer #1 (Remarks to the Author):

This is an interesting manuscript that identifies six main intercalated cell subtypes in human kidneys derived from samples from patients with renal cell carcinoma and other malignancies. Interestingly, upon exposure to uropathogenic *E. coli* the RNAseq profiles of these six cell types showed an increase in gene expression of phagosome maturation pathways. It is encouraging that this group of researchers working with "fresh" human surgical samples and a relatively low sample size have obtained this highly significant data in the kidney research field. The changes upon IC exposure to UPEC are interesting and relevant.

An important point is that the authors do not clearly link or compare findings from the McMahon laboratory in profiling of mice tubular cells to the findings from the human samples presented in this study. This comparison would be very important for the field of intercalated cells in physiology and pathology, as the usual tools for investigating these cells involve animal models. The *in vivo* experiments show technical skill and are convincing.

Major concerns:

1. Given the findings from Chen et al (ref 23) and from Ransick et al (2019) how do the authors compare the current findings to those of mice single cell transcriptomes?
2. Are all IC types expressing FOXI1 compared to non-IC?
3. How do the transcriptomes presented here under the normal conditions relate to the differential male and female changes in gene expression presented in Ransick et al?
4. After UPEC infection, are there any pathways that indicate that the IC are more likely to undergo cell death (by any mechanism), compared to PC?
5. Have the authors attempted to monitor gene expression by IHC for some of the key
6. It would be important to evaluate and clearly report whether there are differential expression changes of defensin genes such as RNA7 and NGAL, and in addition TLR4 in the different IC clusters in the presence or absence of UPEC. These genes are known to be involved in the response of IC to infection.

It is possible that these genes are expressed at different time points after infection compared than the changes in phagocytosis-related genes?

Minor concerns:

1. consider changing the title the term "renal" to "kidney" and throughout the manuscript.
2. consider mentioning in the abstract the term "H⁺-ATPase" to facilitate keyword search, given all the different terminology used for the V-ATPase.
3. Although it is difficult to obtain human kidney samples not from surgical oncology, it would be helpful to show that the six different clusters by marker gene are expressed in human kidney tissue not derived from oncological surgery. This could easily be done via IHC or immunofluorescence labeling and confocal microscopy. Showing expression of these clusters in mice might also be informative.

Reviewer #2 (Remarks to the Author):

This manuscript describes the novel identification of up to 6 distinct IC subtypes and the potential role of ICs in sensing and responding to infection. Specifically, uropathogenic *E. coli* (UPEC) exposure altered the proportion of subtypes, ICs were found to phagocytose UPEC and acidify following phagocytosis, and UPEC infection increased IC ATPase E1 expression in ICs. The contribution of ICs to sensing and responding to pathogens is of high interest to the field and warrants further study, particularly the potential role of the A-IC subtype as an innate immune variant. The manuscript is well-written, statistical analyses appear appropriate, and the conclusions are largely well-supported by the data. Main comments are below.

Based on Supplemental Table 2, the scRNAseq experiment was performed on cells from a single human kidney section. This needs to be more clearly indicated in the main text and limitations section of the discussion.

The enrichment strategy for ICs from human tissue is based on using c-KIT as a surface protein target. However, KIT staining in Supplemental Figure 1 appears to be widespread, but does co-occur with V-ATPase E1 in a small subset of cells. While this limitation is acknowledged in the discussion, it would be helpful to see co-localization of c-KIT and CD45 staining or some other estimation of the fraction of c-KIT⁺ cells that are CD45⁺ vs V-ATPase E1 positive to determine the purity of the IC enrichment. Similarly, is there enrichment for pendrin expression in the resulting cell fraction?

The majority of the data are derived from male tissue/male mice, yet the UTI studies are necessarily performed in female mice. Can the authors speculate on potential sex differences in IC subtypes and response to UPEC?

The text in the majority of the figures is too small to clearly read and often blurry after zooming in sufficiently.

Reviewer #3 (Remarks to the Author):

This study has focused on the individual response of renal intercalated cells (IC) to UPEC. The authors present data from *in vitro* single-cell RNAseq profiling of ICs alone compared to UPEC-exposed ICs and then present *in vivo* work examining co-localization of UPEC with ICs in an "IC-reporter" mouse strain. The major conclusions from the studies is that ICs appear to present macrophage-like properties, by taking up and presumably acidifying the bacterial-containing vacuole.

Major strengths of the Manuscript are the novelty of the work: There are very few studies examining individual cell type interactions of the kidney epithelium with bacterial pathogens. In addition, the authors combine single-cell RNAseq and *intra-vital* microscopy, expanding not only potentially knowledge on host-pathogen interactions, but also expanding the technique toolkit available to the

field.

MAJOR weaknesses to the manuscript are the following:

(1) The manuscript is very poorly and very densely written with substantial and critical information missing or left to be deduced by the readers.

The first sections of the results do not describe how the ICs were collected, and how UPEC was cultured, how long it was cultured for and how the experiment was performed. This is also lacking in the materials and methods. They mention that CFT073 was grown overnight in LB broth but do not specify what multiplicity of infection was used for the IC incubation and for how long were the bacteria in association with the ICs prior to sample collection.

(2) The figures (esp. Figure 1 and S3) are very poor resolution and illegible. This makes it very hard to view the data, let alone interpret the data.

(3) For the intra-vital microscopy, again, it is unclear how bacteria were prepared for infection studies. It is also unclear as to whether the authors coated UPEC strain CFT073 with the pHRODO stain and used that for the uptake analyses, or if they used store-bought E. coli bioparticles. This is critical to clarify, since UPEC strains act quite differently from the K12 E. coli used to coat the bioparticles.

(4) The RNA velocity is not well-described. This is still a fairly new concept and as written in the article it does not provide any useful information to readers who may not be familiar with what this multi-dimensional vector actually means.

(5) Minor - There are multiple typos and wrong words throughout the manuscript that make it even harder to read. For example, in the discussion there is a sentence stating that UPEC phagocytoses UPEC. There are other areas in the manuscript where words and tenses are incorrectly used

We have performed additional experiments and analysis along with revising the manuscript in response to the review panel's insight, concerns and comments. We have extensively revised the text along with adding new figures (Fig 2 and Fig 3) and supplemental material (S2, S6-S10) We appreciate that the reviewers noted the novelty of the work and consistently described their findings as interesting and relevant. Below is an itemized summary of our response to their concerns.

RESPONSE TO REVIEWER #1

Concern 1: An important point is that the authors do not clearly link or compare findings from the McMahon laboratory in profiling of mice tubular cells to the findings from the human samples presented in this study. This comparison would be very important for the field of intercalated cells in physiology and pathology, as the usual tools for investigating these cells involve animal models. The *in vivo* experiments show technical skill and are convincing. Given the findings from Chen et al (ref 23) and from Ransick et al (2019) how do the authors compare the current findings to those of mice single cell transcriptomes?

Response: We have added comparisons of our findings to mouse single cell transcriptomes reported by Ransick *et al* and Chen *et al*. For example, we have added Figure 3 which demonstrates the degree that IC, A-IC, B-IC and PC murine cell type markers reported in the aforementioned expression patterns are conserved in humans. We have also added the following sentences to our discussion:

“The paucity of LCN2/NGAL and TLR4 expression in ICs is consistent to what has been reported in murine ICs by our research group along with Ransick and colleagues (<https://cello.shinyapps.io/kidneycellexplorer/>).^{15, 30} Chen and colleagues did report some TLR4 and NGAL mRNA expression in ICs, but several fold less than in PCs.²³”

“In mice, Ransick and colleagues identified *Slc8a1* expression primarily in “PC-like cells” of the connecting tubule and the distal convoluted tubules.³⁰ On our confocal microscopy evaluating kidney SLC8A1 expression we found isolated cells that expressed SLC8A1 in AQP2 positive tubules consistent with collecting ducts or connecting tubules. We also identified AQP2 negative tubules in which most cells expressed SLC8A1, potentially consistent with the aforementioned distal convoluted tubule SLC8A1 expression in mice”

We also explained why a comparable number of A-IC subtypes were not seen on the prior murine scRNAseq studies by adding the following paragraph to the discussion:

“We performed scRNAseq on 1,861 human kidney cell of which 1,066 were ICs. To contextualize murine experimental models to human pathophysiology, it will be important to compare and contrast human and murine ICs. Chen and colleagues enriched for murine ICs with c-KIT and performed scRNAseq.²³ They stated “74 cells classified as PCs, 87 cells classified as A-ICs, and 23 cells classified as B-ICs”.²³ Ransick and colleagues performed scRNAseq on 688 ICs.³⁰ They did not enrich for ICs but rather divided the kidney into the cortex, inner medulla and outer medulla to evaluate zonal differences.³⁰ We demonstrated that murine collecting duct cell type marker expression appeared to be largely conserved in human collecting duct cells. Whether murine have comparable A-IC subtypes to humans will likely require a targeted single cell evaluation of a larger number of murine ICs.”

Concern 3: Are all IC types expressing FOXI1 compared to non-IC?

Response: Yes, all IC subtypes have high relative expression of FOXI1 compared to non-IC kidney cells. To highlight this finding we have added Figure 3 which contains a dot plot of FOXI1 expression.

Concern 4: How do the transcriptomes presented here under the normal conditions relate to the differential male and female changes in gene expression presented in Ransick et al?

Response: Our scRNAseq and intravital studies both evaluated male ICs. Therefore we do not have female IC data for comparison. We have added the following paragraph to the discussion to address male versus female differences:

“Pyelonephritis pathophysiology has male vs female distinctions. For example, in humans, females are five-times more likely to develop acute pyelonephritis but male mice have androgen mediated increased severity of pyelonephritis.^{66, 67} Ransick and colleagues demonstrated differential male and female differences in the proximal tubule in PCs and proximal tubules, particularly in organic anion transporters.³⁰ Male ICs were evaluated in both our single cell sequencing and intravital studies. Because of the role of ICs in the bacterial defense of the kidney and electrolyte balance, sex-related diversity in IC functions such as phagocytosis and single cell expression will be key areas for future studies”

Concern 6: After UPEC infection, are there any pathways that indicate that the IC are more likely to undergo cell death (by any mechanism), compared to PC?

Because we used a relatively short 1 hour UPEC exposure, our pathway analysis results were likely skewed towards early cell responses to infection. There were no cell death pathways in the top 10 pathways for each cell type that we reported. If we looked at all pathways that the Ingenuity software reported as being significantly involved, we did find that (a) IC Cluster 0 (A IC variant A) had “death receptor signaling pathway” which was downregulated (negative Z-score) in saline treated control and remained downregulated upon UPEC exposure, (b) IC cluster 5 (A IC variant C) had “death receptor signaling pathway” upregulated (positive z-score) in saline control and remained upregulated in UPEC exposed cells, and (c) Cluster 10 (principal cells) had “death receptor signaling pathway” upregulated in saline as well as in UPEC exposed cells. Interestingly this finding is potentially consistent with the data presented on Figure 4 where the percentage of cells in cluster 0 increased while cluster 5 decreased in response to UPEC. We did not report these findings in the revised manuscript because the pathways are fairly far down on the list (>50th). However we did state the following in the discussion:

“ICs phagocytosed bacteria over several minutes, and our cells for scRNAseq were exposed to UPEC for a relatively short 1-hour time point. Future studies are needed to determine if other IC pathways, such as antimicrobial peptide expression or regulation of cell death, become more prominent in later timepoints following UPEC exposure.”

Concern 7: It would be important to evaluate and clearly report whether there are differential expression changes of defensin genes such as RNA7 and NGAL, and in addition TLR4 in the different IC clusters in the presence or absence of UPEC. These genes are known to be

involved in the response of IC to infection. It is possible that these genes are expressed at different time points after infection compared than the changes in phagocytosis-related genes?

Response: We have added supplemental material S6 and S7 that presents expression patterns of select innate immune genes including *TLR4* and *LCN2* (NGAL) in the clusters identified by scRNAseq. We did not identify any *RNASE7* expression in our scRNAseq experiment. For the revision we have performed RT-PCR on bulk sorted ICs from 4 kidney samples and identified *RNASE7* expression in ICs from 1 of 4 patients and in non-ICs in 0 of 4 patients. We have added the following paragraphs to the results and discussion respectively:

“The kidney innate immune profile demonstrates some early single cell expression changes following 1 hr. of UPEC exposure. scRNAseq expression patterns of select innate immune genes following UPEC vs saline exposure for 1 hour are presented in Supplemental material S7 and S8. Key findings include high expression of the antimicrobial peptide adrenomedullin (*ADM*) in A-IC subtypes A and B, hybrid PC-ICs and B-ICs. PCs do not express adrenomedullin at baseline but have significantly increased expression in response to UPEC exposure. Cytokine Inducible SH2 Containing Protein (*CISH*) and Barrier to Autointegration Factor 1 (*BANF1*) expression is significantly induced in nonA nonB ICs following UPEC exposure. Interleukin 18 (*IL18*), galectin 3 (*LGAL3*), beta defensin 1 (*DEFB1*) and signal transducer *CD24*. Lipocalin 2/ neutrophil gelatinase-associated lipocalin (*LCN2/NGAL*) was only identified in hybrid PC-ICs and only minimal toll like receptor 4 mRNA expression was present. We did not identify expression of ribonuclease 7 (*RNASE7*) on scRNAseq. We evaluated *RNASE7* mRNA expression in IC and non-IC kidney cells magnetically sorted into pooled samples and identified mRNA expression in ICs from 1 of 4 kidneys (Supplemental material S9).”

“We identified scRNAseq IC expression of innate immune proteins such as the immune and inflammatory mediator galectin 3 (*LGAL3*) and the antimicrobial peptide adrenomedullin (*ADM*) that we have previously reported in rodent ICs.^{15, 40, 41, 42} However, some key innate immune proteins previously reported in ICs such as, *LCN2/NGAL*, *RNASE7* and *TLR4* had minimal expression or were not seen in human ICs on the single cell level in our analysis.^{6, 9} The paucity of *LCN2/NGAL* and *TLR4* expression in ICs is consistent to what has been reported in murine ICs by our research group along with Ransick and colleagues (<https://cello.shinyapps.io/kidneycellexplorer/>).^{15, 30} Chen and colleagues did report some *TLR4* and *NGAL* mRNA expression in ICs, but several fold less than in PCs.²³ We did evaluate for *RNASE7* expression in pooled human ICs and identified it in ICs from 1 out of 4 kidneys (Supplemental material S9) indicating that its expression may be intermittent depending on region, time point and physiological conditions. *CISH* expression which is induced in nonA-nonB ICs in response to UPEC has been demonstrated to regulate the innate immune response to *M. tuberculosis* in the lung and spleen.⁴³ ICs phagocytosed bacteria over several minutes, and our cells for scRNAseq were exposed to UPEC for a relatively short 1 hour time point. Future studies are needed to determine if other IC pathways, such as antimicrobial peptide expression or regulation of cell death become more prominent in later timepoints following UPEC exposure”

Also see the response to concern 6 regarding timepoints.

Concern 8: consider changing the title the term "renal" to "kidney" and throughout the manuscript.

Response: We have changed the term “renal” to “kidney” throughout the manuscript with the exception of limited situations where the term “renal” would be standard such as in “renal tubular acidosis”.

Concern 9: consider mentioning in the abstract the term "H⁺-ATPase" to facilitate keyword search, given all the different terminology used for the V-ATPase.

Response: We have added the following statement to the abstract: “Intercalated cells are involved in acid base homeostasis via vacuolar ATPase (H⁺-ATPase or V-ATPase) expression”

Concern 10: Although it is difficult to obtain human kidney samples not from surgical oncology, it would be helpful to show that the six different clusters by marker gene are expressed in human kidney tissue not derived from oncological surgery. This could easily be done via IHC or immunofluorescence labeling and confocal microscopy. Showing expression of these clusters in mice might also be informative.

Response: To address this question, we performed triple immunolabeling confocal microscopy with SLC8A1, AQP2 and V-ATPase E1 and identified hybrid PC-ICs to demonstrate that protein expression and cellular localization was consistent with the scRNAseq results. We also performed triple labeling with EGR1, HSPA1A and V-ATPase E1 to demonstrate that that protein expression and cellular localization was consistent with scRNAseq results for IC subtypes. The results are presented in Figure 2. The only kidney tissue we have access to is the normal margins of mass resections. We did have one kidney sample that the underlying condition was focal segmental glomerulosclerosis (FSGS) not a malignancy. We used this kidney for RT-PCR only because we receive the pathology report several weeks after the tissue. To confirm that our findings were not artifact of the elderly patients with kidney cell carcinomas, we confirmed that the immunohistochemistry of SLC8A1, HSPA1A and EGR1 in normal kidney tissue as identified in the Human Protein Atlas (<http://www.proteinatlas.org>) (Supplemental material S6) was consistent with our confocal microscopy results. Please note that we have replaced FOSB and ABCA5 with HSPA1A and EGR1 as A-IC subtype markers because HSPA1A and EGR1 immunolabeling in the kidney resulted in much better images than FOSB and EGR1. Additionally the pairs had comparable IC subtype expression profiles.

RESPONSE TO REVIEWER #2

Concern 11: Based on Supplemental Table 2, the scRNAseq experiment was performed on cells from a single human kidney section. This needs to be more clearly indicated in the main text and limitations section of the discussion.

Response: We have clarified that the scRNAseq experiment was performed on cells from a single human kidney sample in the results and methods.

“The enriched ICs were obtained from the normal margins of a single kidney mass resection.”

“The UPEC exposure to human ICs in our scRNAseq was *in vitro* to a single kidney sample derived cells”

Concern 12: The enrichment strategy for ICs from human tissue is based on using c-KIT as a surface protein target. However, KIT staining in Supplemental Figure 1 appears to be widespread, but does co-occur with V-ATPase E1 in a small subset of cells. While this limitation is acknowledged in the discussion, it would be helpful to see co-localization of c-KIT and CD45 staining or some other estimation of the fraction of c-KIT⁺ cells that are CD45⁺ vs V-ATPase E1 positive to determine the purity of the IC enrichment. Similarly, is there enrichment for pendrin expression in the resulting cell fraction?

We have clarified the methodology to highlight that we magnetically removed CD45+ cells before we magnetically sorted ICs from non-ICs by adding the following sentence to the methodology:

“Live cells were counted and CD45⁺ immune cells were removed using anti-Human CD45 micro beads (Cat. No. 130-045-801, Miltenyi Biotec) before the ICs were enriched”

Additionally we have performed additional bulk magnetic sorting of ICs vs non-ICs and used Rt-PCR to demonstrate relative enrichment of ATP6V1B1 (ICs), SLC4A1 (A-ICs) and SLC26A4 (B-ICs) and presented this data as Supplemental material S2. We did enrich for pendrin in some of the human kidney samples. This highlights that there may regional or individual differences in kidney cell type concentrations.

We further reviewed the literature on c-KIT/ expression in myeloid/ hematopoietic cells and in the kidney. C-KIT is expressed, but in progenitor cells which are not likely to represent kidney resident immune cell population. Rusu et al identified C-KIT by IHC in kidney stromal cells/telocytes which we did not see in our scRNAseq results. Miliaras et al did identify some isolated proximal tubule and loop of Henle positivity which is consistent with our RNAseq results.

We have revised the discussion to state: “ICs comprised 57% of the cells in our scRNAseq, enriched from the ~7% IC composition in kidneys at baseline.³³ Our sample did contain some other epithelial and endothelial cell types but was negative for CD45+ immune cells which were sorted off prior to enrichment of ICs (Figure 1, Supplemental material S1). With scRNAseq, the target ICs could be identified and evaluated separately. Hematopoietic stem cells express c-Kit but they do not likely to represent the kidneys resident myeloid cells populations.⁶⁸ We did not identify other c-KIT expressing cells, present in health human kidneys such as stromal cells/telocytes.⁶⁹ Some isolated loop of Henle and proximal tubule c-KIT positivity has been previously reported on kidney immunohistochemistry consistent with our scRNAseq results.⁷⁰ Additionally some green autofluorescence by red blood cells during IF imaging can occur”

We have revised the legend for supplemental material S1 to include: “Of note c-KIT also stains stromal cells/telocytes and can intermittently stains some proximal tubules and loop of Henle cells. These aforementioned cell types may represents the c-KIT staining that did not localize with the V-ATPase E1 subunits.”

Concern 13: The majority of the data are derived from male tissue/male mice, yet the UTI studies are necessarily performed in female mice. Can the authors speculate on potential sex differences in IC subtypes and response to UPEC?

Response: We have addressed this as outlined in the response to concern 4

Concern 14: The text in the majority of the figures is too small to clearly read and often blurry after zooming in sufficiently.

Response: We have uploaded each figure and figure legends individually to address this barrier to optimal visualization. Additionally we have increased the resolution to at least 300 dpi.

RESPONSE TO REVIEWER #3

Concern 15: The manuscript is very poorly and very densely written with substantial and critical information missing or left to be deduced by the readers. The first sections of the results do not describe how the ICs were collected, and how UPEC was cultured, how long it was cultured for and how the experiment was performed. This is also lacking in the materials and methods. They mention that CFT073 was grown overnight in LB broth but do not specify what multiplicity of infection was used for the IC incubation and for how long were the bacteria in association with the ICs prior to sample collection.

Response: We have expanded our description of this methodology and added a dedicated paragraph to the methods to clarify how the UPEC was handled and ICs were obtained.

“Human kidney tissue, primarily from normal margins of kidney mass resections was dissected into 2-4 mm pieces then overnighted to our lab in Dulbecco's Modified Eagle Medium (DMEM) on ice by the Cooperative Human Tissue Network (<https://www.chtn.org>).”

“ScRNA-seq was completed on 1,861 dissociated kidney cells that were enriched for ICs as described above”

“*In vitro* exposure of dissociated kidney cells to UPEC and saline UPEC were grown overnight in Luria broth at 300 rpm and 37°C. An aliquot of bacterial broth was pelleted and suspended in sterile PBS. Optical density of the culture was measured at 600 nm. For *in vitro* stimulation Based on this calculation; OD600 of 1=8x10⁸ cells/ml, 1x10⁵ bacterial cells were estimated (in ~20 µl bacterial suspension). Enriched IC cells as prepared above were equally divided in 2 wells of 96 well u-bottom plate and incubated for 1 hr. at 37°C and 5%CO₂ in ~180 µl (total volume 200 µl) DMEM containing 10% FBS with 1x10⁵ UPEC cells or equal volume of sterile PBS alone. After incubation, cells were centrifuged at 1200 rpm for 10 min and media was carefully removed and cells were washed again with sterile PBS, suspended in 50 µl PBS (without Ca²⁺ or Mg²⁺) and were taken to the IUSM genomics core for 10x chromium single cells preparation and sequencing.

We have also added the following sentence to the discussion:

“Future directions using different bacterial strains and growth conditions will lead to increased understanding of bacteria-IC interactions. For example, the pyelonephritis strain that we used (UPEC CFT073) could be compared to a cystitis strain such as UPEC strain UTI-89 or the *E.coli* K12 strain is generally used to coat bioparticles.⁶⁹ Additionally

different bacterial growth conditions could be compared, for example UTI-89 has been demonstrated to have increase type 1 pili formation if grown statically.⁷⁰

Concern 16: The figures (esp. Figure 1 and S3) are very poor resolution and ineligible. This makes it very hard to view the data, let alone interpret the data.

We included the figures in the word document originally. For the revision, we have uploaded figures individually and increased DPI to at 300 which should optimize resolution, particularly for detailed figures such as Figure 1. We turned Figure S3 into a multipage supplemental page to improve visualization.

Concern 17: For the intra-vital microscopy, again, it is unclear how bacteria were prepared for infection studies. It is also unclear as to whether the authors coated UPEC strain CFT073 with the pHRODO stain and used that for the uptake analyses, or if they used store-bought *E. coli* bioparticles. This is critical to clarify, since UPEC strains act quite differently from the K12 *E. coli* used to coat the bioparticles.

We have clarified that bacteria were prepared as previously described for the scRNAseq experiment. We also clarified that we used commercial *E. coli* bioparticles

“Commercially obtained *E. coli* bioparticles (pHrodo Green Bioparticles, Cat no. P35381, Molecular probes, Invitrogen) or GFP-UPEC strain CFT073 were injected into a collecting duct”

Concern 18: The RNA velocity is not well-described. This is still a fairly new concept and as written in the article it does not provide any useful information to readers who may not be familiar with what this multi-dimensional vector actually means.

We have added the following explanation regarding RNA velocity:

The relative abundance of recently transcribed unspliced pre-mRNAs versus mature spliced mRNA can be used to calculate the changes in mRNA abundance, termed mRNA velocity (represented by arrow vector).³¹ A positive mRNA velocity (increased arrow size) in single cells studies indicates that genes are being upregulated where a negative mRNA velocity (reduced arrow size) means that genes are being downregulated.³¹ The RNA velocity direction infers a cell has a mRNA expression trajectory toward another cell type.^{31, 32}

Concern 19: Minor - There are multiple typos and wrong words throughout the manuscript that make it even harder to read. For example, in the discussion there is a sentence stating that UPEC phagocytoses UPEC. There are other areas in the manuscript where words and tenses are incorrectly used

We have extensively edited the manuscript correcting the wording and tenses.

REVIEWERS' COMMENTS

Reviewer #1 (Remarks to the Author):

I thank the reviewers for addressing my concerns. The authors have significantly clarified the methods, including the data analyses. Their responses and added text put their work in a better context with the findings from known, more accessible IC study platforms such as murine models. The manuscript in its revised form is a tour de force article that will likely inform not only changes in IC characteristics in the setting of infection but also normal IC physiology.

Reviewer #2 (Remarks to the Author):

The revised manuscript has sufficiently addressed all prior reviewer concerns. Minor points remaining to be addressed are below:

Figure 6 is difficult to follow. Consider maintaining the order of the pathways and adding a "Rank" designation to the right of the graph.

The sentence in lines 229-230 is unclear.

The reference to Supplemental material S11 in line 243 seems out of place, as this figure is a diagram of phagolysosomal maturation rather than the implied comparison of percentage of ICs with green bioparticles.

The reference to Supplemental material S12 in lines 247-248 is incorrect, as this figure reports flow cytometry data rather than pathway analyses.

Line 356: please clarify "differences in the proximal tubule in PCs and proximal tubules"

Line 366: "form" should be "from"

Reviewer #3 (Remarks to the Author):

The authors have adequately addressed the reviewer's concerns and have greatly improved the clarity of the manuscript. This work is significant and no further concerns or suggestions are raised.

We have made revisions to address the reviewer's comments and also made some minor grammar, spelling and punctuation corrections in track changes. Below is an itemized response to the reviewer's concerns

Concern 1: Figure 6 is difficult to follow. Consider maintaining the order of the pathways and adding a "Rank" designation to the right of the graph.

Response: we have added a Rank designation to the graph as recommended

Concern 2: The sentence in lines 229-230 is unclear.

Response: We have revised the wording on line 229-230 from: "We repeated this intravital tubular perfusion experiment using pHrodo coated *E. coli* BioParticles™ for Phagocytosis which only fluoresce when acidified. tdTomato positive (tdT⁺), a red fluorescence protein variant expression ICs, but not tdT⁻ presumed PCs fluoresced (**Figure 8**)."

to "We also perfused tubules with pHrodo coated *E. coli* BioParticles™ for Phagocytosis which only fluoresce when acidified. Uptake and fluorescence of these bioparticles was only visualized during intravital microscopy in ICs (**Figure 8**)."

Concern 3: The reference to Supplemental material S11 in line 243 seems out of place, as this figure is a diagram of phagolysosomal maturation rather than the implied comparison of percentage of ICs with green bioparticles, The reference to Supplemental material S12 in lines 247-248 is incorrect, as this figure reports flow cytometry data rather than pathway analyses.

Response: The legends for S11 and 12 were reversed and have been corrected

Concern 4: Line 356: please clarify "differences in the proximal tubule in PCs and proximal tubules"

Response: we have corrected "differences in the proximal tubule in PCs and proximal tubules" to "differences in proximal tubule cells"

Concern 5: Line 366: "form" should be "from"

Response: "form" was corrected to "from"